# GENERALIZATION GRADIENT DESCENT

## ABSTRACT

We propose a new framework for evaluating the relationship between features and generalization via a theoretical analysis of the out-of-distribution (OOD) generalization problem, in which we simultaneously use two mathematical methods: a generalization ratio that quantitatively characterizes the degree of generalization, and a generalization decision process (GDP) that formalizes the relationship of loss between seen and unseen domains. By combining the concepts of informativeness and variation in the generalization ratio, we intuitively associate them with OOD problems to derive the generalization inequality. We then introduce it to the GDP to select the best loss from seen domains to gradient descent for backpropagation. In the case where the classifier is defined by fully connected neural network, the entire system is trained with backpropagation. There is no need for any model selection criterion or operating on gradients during training. Experiments demonstrate the potential of the framework through qualitative and quantitative evaluation of the generalization ability.

## 1 INTRODUCTION

Traditional supervised learning is highly dependent on the proposition of independent and identically distributed (i.i.d.), while ignoring out-of-distribution (OOD) scenarios commonly encountered in real-word applications (Taori et al., 2020; Zhou et al., 2022; Wang et al., 2022; Yang et al., 2021; Wang et al., 2024a). So far, research on the success of gradient operations in generalization algorithms has emerged (Huang et al., 2020; Shi et al., 2021; Rame et al., 2022; Tian et al., 2022; Wang et al., 2021), generally trying to learn generalized representations by directly operating on gradients. These successes have primarily been based on the manipulation and transformation of features. However, theoretical analysis for the OOD generalization problem has less impact, due to the difficulty of providing a metric for evaluating the relationship between features and generalization when features undergo shifts, and due to the complexity in integrating adjustments for generalization into the training procedure. We propose a novel quantitative metric called "generalization ratio" and a new estimation procedure called "generalization decision process" to sidestep these difficulties.

In the proposed generalization gradient descent (GGD) framework, an available domain set $\mathcal{E}_{avail}$ includes both the training set $\mathcal{E}_{tra}$ and the validation set $\mathcal{E}_{val}$. The quantitative metric evaluates the degree of generalization between $\mathcal{E}_{tra}$ and $\mathcal{E}_{avail}$. Our aim is to utilize $\mathcal{E}_{tra}$ to generalize to $\mathcal{E}_{val}$, and then achieve generalization to a larger domain set $\mathcal{E}_{avail}$, which includes all unseen domains. In other words, we consider the validation set as an unseen domain, which is not utilized for backpropagation during training. Firstly, we define ideal feature matrix function, generalization and non-generalization models (Definition 3.1, 3.2 and 3.3), where we refer to the definition of generalization proposed in the previous theoretical framework (See Ye et al. (2021)). Considering the actual distribution and handling of features in gradient descent, we further clarify and redefine these definitions, i.e., we define the "variation" and "informativeness" (Definition 3.4 and 3.5) of features for classification task, and propose propositions (Proposition3.6 and 3.7). Meanwhile, to incorporate the definitions of "variation" and "informativeness" of features, we quantitatively define the generalization ratio (Definition 3.8 and Theorem 3.9) based on the definition of expansion function (Definition 3.3 in Ye et al. (2021)). In other words, the generalization ratio is a kind of metric to quantitatively characterize the degree of generalization. Then, through the definition of learnability (Definition 3.4 in Ye et al. (2021)), the generalization ratio is also a learnable OOD problem (Theorem 3.10). In our experiment, we will introduce easily computable definitions of "variation" and "informativeness" through Bayes' theorem for the estimation procedure (Theorem 2.1). As pointed out by previous

theoretical framework (See Ye et al. (2021)), any high-dimensional joint distributions where the marginal distributions in each dimension are close but non-overlapping. To address this challenge, we theoretically confirm that even with variations in distribution, it is still possible to accurately predict the $\mathcal{E}_{avail}$ dataset using a generalized model (Theorem 4.1), consistent with the definition of generalized model (Definition 3.3). Meanwhile, we prove the generalization inequality (Theorem 4.2) based on the generalization ratio and the linear top model theorem (Theorem 4.2 in Ye et al. (2021)), formalizing the relationship between generalization error and loss on $\mathcal{E}_{tra}$ and $\mathcal{E}_{avail}$. Specifically, we consider both nonlinear and linear neural networks to establish the structure of the model. Based on these models, we derive a generalization inequality to create an estimation procedure.

To combine the generalization inequality with gradient descent in an algorithm, inspired by reinforcement learning, we introduce the generalization decision process (GDP) as an estimation procedure. In this process, we use the generalization inequality as the basis, defining deterministic transitions (Definition 5.1 and 5.2) for both generalization and non-generalization based on changes in both generalization error and loss on $\mathcal{E}_{tra}$, while providing various magnitudes of rewards. We consider the generalization ratio as the state and the losses on $\mathcal{E}_{tra}$ as the actions. This aims to find the maximum rewards within the deterministic transitions to select the best action, and then utilizes backpropagation with gradient descent to optimize this action. In addition, we use $T$ as the stopping criterion of iterations, allowing the process to stop at any time (not necessarily until total convergence). This framework can provide specific training algorithms for all types of neural network and optimization algorithm. In this article, we utilize quantitative metrics during model training in simple cases use the GDP for backpropagation with gradient descent.

**Contribution.** In summary, this paper will provide the following: (1) We introduce a quantitative metric called "generalization ratio" that characterizes the relation between $\mathcal{E}_{tra}$ and $\mathcal{E}_{avail}$ in terms of the variation and informativeness of features. (2) We theoretically prove the generalization inequality in terms of generalization ratio and linear top model theorem. This inequality intuitively guarantees an upper bound on the loss on $\mathcal{E}_{avail}$, which consists of both seen and unseen domains. (3) We theoretically confirm that accurate predictions remain possible even in the case of distributional variations. (4) We propose an estimation procedure called the "Generalization Decision Process", a reinforcement learning approach to address the combined problem in the gradient descent algorithm.

## 2 RELATED WORKS

Domain generalization (Zhou et al., 2022) and OOD generalization (Yu et al., 2024; Yang et al., 2024) have garnered significant attention in recent years, with extensive research addressing the challenge of domain shift. Domain adaptation (DA) (Li et al., 2024; Fang et al., 2024), in particular, has been a focal point of this literature, with numerous approaches proposed to mitigate the discrepancies between the training (source) and the test (target) distributions (Chen, 2024; Moreno-Torres et al., 2012; Recht et al., 2019; Ben-David et al., 2010; Taori et al., 2020; Blanchard et al., 2021). Since the formal introduction of domain generalization in 2011 by Blanchard et al. (2011), several methods have been developed to address the generalization of OOD, addressing the problem of distribution change through various strategies (Yang et al., 2024; Teney et al., 2024; Zhou et al., 2021; Fan et al., 2021; Zhang et al., 2021; Pandey et al., 2021; Shu et al., 2021; Zhou et al., 2024; 2023; Cha et al., 2021).

Many works have emerged from the perspectives of causal discovery, distributional robustness, and conditional independence, with the aim of providing robust solutions to the OOD generalization problem (Gui et al., 2024; Wang et al., 2024b; Ahuja et al., 2020a; Bai et al., 2021; Creager et al., 2020; Chang et al., 2020; Jin et al., 2020; Koyama & Yamaguchi, 2020; Krueger et al., 2021; Parascandolo et al., 2020; Sagawa et al., 2019; Xie et al., 2020). These approaches often focus on defining test domains around the training domain using distribution distance measures or using causal frameworks that remain invariant under interventions (Shao et al., 2024; Arjovsky et al., 2019; Heinze-Deml & Meinshausen, 2021; Magliacane et al., 2018; Meinshausen, 2018; Müller et al., 2021; Pfister et al., 2021; Rojas-Carulla et al., 2018; Schölkopf et al., 2012). The principle underlying these methods is that a causal model, which achieves minimal worst-case risk, is invariant to distribution shifts (Wu et al., 2024; Aldrich, 1989; Haavelmo, 1944; Pearl, 2009; Rojas-Carulla et al., 2018). However, the unknown nature of test distributions necessitates additional assumptions for effective generalization analysis.

Critically, some studies have highlighted the limitations of existing methods from both theoretical and experimental perspectives, highlighting areas where current approaches fall short (Yu et al., 2024; Feder et al., 2024; Ahuja et al., 2020b; Gulrajani & Lopez-Paz, 2020; Kamath et al., 2021; Nagarajan et al., 2020; Rosenfeld et al., 2020). Recently, methods utilizing gradient information to enforce generalized representations (Wang et al., 2022) and reinforcement learning-based techniques that train agents to generalize to target environments (Ye et al., 2023) have been introduced, expanding the scope of solutions beyond the environmental-focused strategies of traditional targets. For more comprehensive discussions on domain adaptation and OOD generalization, including recent advances, please refer to see Yang et al. (2024); Wang et al. (2022); Zhou et al. (2022); Yang et al. (2021).

The rest of the section is as follows: Section 3 is our preliminary. We give our quantitative metric of the generalization ratio in Section 4 and the generalization inequality in Section 5. We propose the generalization decision process (GDP) in Section6 and the generalization gradient descent (GGD) algorithm in Section 7. We conduct our experiments and discuss further limitations and future directions in Section 8, review related works in Section 2, and conclude in Section 9.

## 3  PRELIMINARY

We consider a multi-class task $\mathcal{X} \rightarrow \mathcal{Y} = \{1, \ldots, K\}$. Let $\mathcal{E}_{tra}$ be the training set used during the training procedure, and $\mathcal{E}_{val}$ be the validation set not used for backpropagation during the training procedure, where $\mathcal{E}_{val} \bigcap \mathcal{E}_{tra} = \phi$. The available set we ultimately want to generalize to is $\mathcal{E}_{avail} = \mathcal{E}_{val} \dot{\cup} \mathcal{E}_{tra}$. In our paper, we denote $(X^e, Y^e)$ to be the input-label pair drawn from the data distribution of domain $e$, and assume that $\forall e_i, e_j \in \mathcal{E}$ where $e_i \neq e_j$, they are distinct domains. The goal of risk minimization is to learn a prediction function $f(x)$ that minimizes the expected loss, i.e. $\min_f \mathbb{E}\left[\ell\left(f\left(X^e\right), Y^e\right)\right]$, where $\ell(\cdot, \cdot)$ is a loss function (e.g., the cross-entropy loss) that measure output probabilities between classes and $f$ is a classifier. Integrating the goal of out-of-distribution (OOD) generalization, we find a classifier $f^*$ that minimizes the worst-domain loss on $\mathcal{E}_{avail}$:

$$f^* = \arg\min_{f \in F} \max_{e \in \mathcal{E}_{avail}} \mathcal{L}(e, f), \mathcal{L}(e, f) \triangleq \mathbb{E}\left[\ell\left(f\left(X^e\right), Y^e\right)\right] \tag{1}$$

where $\mathcal{F} : \mathcal{X} \rightarrow \mathbb{R}^K$ is a hypothetical space. Throughout the paper, we consider the nonlinear neural network as the base model $h$ and the linear neural network as the classifier $f$. Let $H$ represent the set of all types of nonlinear neural networks, and $F$ represent the set of all types of linear neural networks. With previous work, $f \in F$ can similar to be decomposed into $g \circ w$, where $g \in \mathcal{G} : \mathbb{R}^n \rightarrow \mathbb{R}^K$ is the top classifier, and $w \in \mathcal{W} : \mathcal{X} \rightarrow \mathbb{R}^n$ is a n-dimensional feature extractor. Based on the linear neural network composed of linear layers, given any $i$-th linear layer, we can decompose $f$ into $g_i \circ w_i$, where $g_i$ is the top linear classifier, and $w_i$ is the linear feature extractor. We denote $W$ as a feature matrix function containing all linear feature extractors decomposed in $f$, where $w_{ij} \in W$ represents a feature map on the $j$-th dimension in the $i$-th layer. Additionally, $W$ is also a function of random variable $X$, obtaining all features of $f$, i.e.,

$$W(X) = [w_{ij}(X)]_{d \times N}, N = max\{n_i\}_{i=1}^d \tag{2}$$

where each $w_{ij} \in W$ map $X$ to $\mathbb{R}^N$ and $d$ represents the total number of layers. The maximum dimension $N$ is found across all layers, and any deficient dimensions are padded with zeros to increase all dimensions to $N$. Then, given a domain $e \in \mathcal{E}$ we denote the feature matrix function as $W(X^e)$, and the conditional distribution of $W(X^e)$ given $Y^e = y$ as $p(W(X^e)|y)$. Our framework can use both balanced and imbalance data. For simplicity, we assume that the data distribution in any domain is balanced, i.e., $\forall y \in \mathcal{Y}, e \in \mathcal{E}$, we have $P(Y^e = y) = \frac{1}{K}$. In our experiments, we transform the conditional distribution of $W(X^e)$ given $Y^e = y$ into experimental form $p(W(X^e)|y)$ using Bayes' theorem. The following theorem presents the conditional distribution:

**Theorem 2.1** (Experimental Form of Features). *Let $e_1, \ldots, e_k \in \mathcal{E}$ be domains such that $P(e_i) > 0$, $i = 1, 2, \ldots, k$. Assume further that $e_1, \ldots, e_k$ also form a partition of the domain $\mathcal{E}$, and $\forall e_i, e_j \in \mathcal{E}$ where $e_i \neq e_j$, they are distinct domains. Let $y$ be any class in $Y$ and $W$ be a feature matrix function. Then we have the conditional probability of features given a class as follows:*

$$\mathbb{P}(W(X^{e_i})|y) = \frac{P(W(X^{e_i})) P(y|W(X^{e_i}))}{\sum_{i=1}^k (P(W(X^{e_i})) P(y|W(X^{e_i})))}. \tag{3}$$

The detailed proof of Theorem 2.1 is included in the Appendix.

# 4 QUANTITATIVE METRIC OF GENERALIZATION RATIO

We first define the "ideal feature matrix function", then introduce the concepts of "generalized model" and "non-generalized model", which mathematically describe the generalized states of a model. Specifically, we use the KL-divergence as the distance measure in the following paper. The non-generalized model and generalized model are defined as follows:

**Definition 3.1** (Ideal Feature Matrix Function). *A feature matrix function $W$ is called an ideal feature matrix function if there exist $e' \in \mathcal{E}_{val}$ and $e'' \in \mathcal{E}_{tra}$, such that $\forall y \in \mathcal{Y}$, we have $P\left(y\middle|W\left(X^{e'}\right)\right) = P\left(y\middle|W\left(X^{e''}\right)\right)$, where $W \in \mathcal{W}^*$ and $\mathcal{W}^*$ is the set of ideal feature matrix functions. If $\exists y \in \mathcal{Y}$ such that $P\left(y\middle|W\left(X^{e'}\right)\right) \neq P\left(y\middle|W\left(X^{e''}\right)\right)$, then $W$ is called a non-ideal feature matrix function, where $W \notin \mathcal{W}^*$.*

**Definition 3.2** (Non-generalized Model). *We say the classifier $f \in F$ is a non-generalized model if there exists a non-ideal feature matrix function $W \notin \mathcal{W}^*$, such that: $\forall e' \in \mathcal{E}_{val}$, $\forall e'' \in \mathcal{E}_{tra}$, $\exists y \in \mathcal{Y}$, we have $P\left(y\middle|W\left(X^{e'}\right)\right) \neq P\left(y\middle|W\left(X^{e''}\right)\right)$ and $P\left(y\middle|W\left(X^{e'}\right)\right) \neq P\left(y\middle|W^*\left(X^{e'}\right)\right)$, where $W^* \in \mathcal{W}^*$.*

**Definition 3.3** (Generalized Model). *We say the classifier $f \in F$ is a generalized model if there exists an ideal feature matrix function $W \in \mathcal{W}^*$, such that: $\forall e' \in \mathcal{E}_{val}$, $\forall e'' \in \mathcal{E}_{tra}$, $\forall y \in \mathcal{Y}$, we have $P\left(y\middle|W\left(X^{e'}\right)\right) = P\left(y\middle|W\left(X^{e''}\right)\right) = P\left(y\middle|W^*\left(X^{e'}\right)\right)$, where $W^* \neq W$ and $W^* \in \mathcal{W}^*$.*

Under balanced data, regardless of which ideal feature matrix function and input $X^e$ are used, the distribution of the output $y$ remains unchanged. Hence, the generalized model ensures that the output distribution of $y$ remains consistent when the different feature matrix functions can be represented by the ideal feature matrix functions. In contrast, the output $y$ of the non-generalized model has a different output distribution. Therefore, we use this characteristic to optimize the non-generalized model and converge it towards a generalized model.

To achieve domain generalization across the available domains, we redefine the definitions in Ye et al. (2021) and utilize the feature matrix function to encompass all the features in the model. Regardless of the variation and informativeness in the domain, the model can effectively generalize to $\mathcal{E}_{val}$. The definitions are as follows:

**Definition 3.4** (Variation). *The variation of the set of scalar maps $W(\cdot)$ from the training set $\mathcal{E}_{tra}$ to the validation set $\mathcal{E}_{val}$ is*

$$\mathcal{V}_{KL}\left(W, \mathcal{E}_{avail}\right) = \max_{y \in \mathcal{Y}} \sup_{\substack{e' \in \mathcal{E}_{val}, \\ e'' \in \mathcal{E}_{tra}}} KL\left(\mathbb{P}\left(W\left(X^{e'}\right)\middle|y\right), \mathbb{P}\left(W\left(X^{e''}\right)\middle|y\right)\right), \quad (4)$$

*where KL is KL-divergence and we concern the set relation rather than the method used for splitting.*

**Definition 3.5** (Informativeness). *The informativeness of the set of scalar maps $W(\cdot)$ across the training set $\mathcal{E}_{tra}$ is*

$$\mathcal{I}_{KL}\left(W, \mathcal{E}_{tra}\right) = \frac{1}{n(n-1)}\left(\sum_{\substack{y \neq y' \\ y, y' \in \mathcal{Y}}} \min_{e'' \in \mathcal{E}_{tra}} KL\left(\mathbb{P}\left(W\left(X^{e''}\right)\middle|y\right), \mathbb{P}\left(W\left(X^{e''}\right)\middle|y'\right)\right)\right) + \bar{\epsilon}, \quad (5)$$

*where $\bar{\epsilon} > 0$ is approximately equal to zero positive value, and KL is KL-divergence.*

The variation $\mathcal{V}_{KL}\left(W, \mathcal{E}_{avail}\right)$ measures the features in the feature matrix function that we expect to preserve in $\mathcal{E}_{tra}$. The informativeness $\mathcal{I}_{KL}\left(W, \mathcal{E}_{tra}\right)$ measures the features in the feature matrix function that we expect to use in $\mathcal{E}_{tra}$. The calculation of informativeness is based on $\mathcal{E}_{tra}$, aiming for low variance while maintaining a certain amount of informativeness. To avoid misclassifying a non-generalized model as a generalized model, we state our propositions as follows:

**Proposition 3.6** *Suppose $\mathcal{E}_{val} \neq \phi$ and $\mathcal{E}_{tra} \neq \phi$, where $\mathcal{E}_{val}$ and $\mathcal{E}_{tra}$ are not i.i.d. If $\mathcal{E}_{val}$ is the domain set we want to generalize to, such that $\forall e' \in \mathcal{E}_{avail}$, $\forall e'' \in \mathcal{E}_{tra}$, and $\exists y \in \mathcal{Y}$, where $p\left(y\middle|W\left(X^{e'}\right)\right) \neq p\left(y\middle|W\left(X^{e''}\right)\right)$, then $KL\left(\mathbb{P}\left(W\left(X^{e'}\right)\middle|y\right), \mathbb{P}\left(W\left(X^{e''}\right)\middle|y\right)\right) \neq 0$.*

**Proposition 3.7** *Suppose the conditions of $\mathcal{E}_{val}$ and $\mathcal{E}_{tra}$ in Proposition 3.6. hold. If we can get a generalized model $f \in F$ such that $\forall e' \in \mathcal{E}_{val}$, $\forall e'' \in \mathcal{E}_{tra}$, and $\forall y \in \mathcal{Y}$, we have $p\left(y\middle|W\left(X^{e'}\right)\right) = p\left(y\middle|W\left(X^{e''}\right)\right)$, then it follows that $KL\left(\mathbb{P}\left(W\left(X^{e'}\right)\middle|y\right), \mathbb{P}\left(W\left(X^{e''}\right)\middle|y\right)\right) = 0$.*

Proposition3.6 states that the variation $\mathcal{V}_{KL}(W, \mathcal{E}_{avail})$ in a non-generalized model requires non-empty and not i.i.d. domain sets as a condition. In contrast, Proposition3.7 states that if the condition in Proposition3.6 is hold, we can find a generalized model with no variation $\mathcal{V}_{KL}(W, \mathcal{E}_{avail})$ between the outputs on $\mathcal{E}_{tra}$ and $\mathcal{E}_{val}$. The detailed proof of propositions is included in the Appendix.

Now we introduce the quantitative metric "generalization ratio". We define the following:

**Definition 3.8** (Generalization Ratio). *The generalization capability of classifier f is*

$$GR_{KL}(W, \mathcal{E}_{avail}) = \frac{\mathcal{V}_{KL}(W, \mathcal{E}_{avail})(\mathcal{I}_{KL}(W, \mathcal{E}_{tra}) + 1)}{\mathcal{I}_{KL}(W, \mathcal{E}_{tra})}, \tag{6}$$

*where $GR(\cdot) > \mathcal{V}(\cdot)$ and $GR(\cdot)$ follow properties hold: 1) $GR(\cdot)$ is monotonically increasing and $GR(\cdot) \geq \mathcal{V}(\cdot), \forall \mathcal{V}(\cdot) \geq 0$; 2) $\lim_{\mathcal{V}(\cdot) \to 0^+} GR(\cdot) = 0$ (We omit the subscript KL in case of no ambiguity).*

**Theorem 3.9** (Expansion Function). *The generalization ratio is an expansion function (Definition 3.3 in Ye et al. (2021)).*

Generalization ratio uses the variation between $\mathcal{E}_{tra}$ and $\mathcal{E}_{val}$, along with the informativeness, to adjust the variation. We combine the variation $\mathcal{V}_{KL}(W, \mathcal{E}_{avail})$ and the informativeness $\mathcal{I}_{KL}(W, \mathcal{E}_{tra})$, satisfying the properties of the expansion function (Definition 3.3 in Ye et al. (2021)) to quantitatively measure the generalized states and guarantee OOD generalization. We give our proof in the Appendix.

Furthermore, generalization ratio is also a $(s(\cdot), \delta)$-learnable OOD problem, as defined in Ye et al. (2021), intuitively associating them with OOD problems. This ensures that the generalization ratio quantifies the generalized states, ensuring that when minimizing variation, the importance of maintaining a certain level of informativeness is not compromised. The following theorem demonstrates that the generalization ratio is also a $(s(\cdot), \delta)$-learnable OOD problem:

**Theorem 3.10** (Learnability on Non-generalized Model). *Let $W$ be a feature matrix function of the non-generalized model $f \in F$ defined in 3.2. Assuming $\mathcal{E}_{avail}$ to be the available domain set and $\mathcal{E}_{tra} = \mathcal{E}_{avail} \backslash \mathcal{E}_{val}$ to be the training set, we say that the non-generalized model is learnable if there exists a generalization ratio $GR_{KL}(W, \mathcal{E}_{avail})$ and $\bar{\epsilon} > 0$, such that: given $W$ satisfy $\mathcal{I}_{KL}(W, \mathcal{E}_{tra}) \geq \bar{\epsilon}$, we have $GR_{KL}(W, \mathcal{E}_{avail}) > \mathcal{V}_{KL}(W, \mathcal{E}_{avail})$. If such the generalization ratio $GR_{KL}(W, \mathcal{E}_{avail})$ and $\bar{\epsilon}$ exist, we also call this model $(GR_{KL}(W, \mathcal{E}_{avail}), \bar{\epsilon})$-learnable.*

We prove that the generalization ratio is a learnable OOD problem through the definition of learnability (Definition 3.4 in Ye et al. (2021)), and demonstrate that under the framework condition, the generalization ratio can serve as a learnability criterion for the OOD problem. Specifically, the statement of this theorem introduces the Definition 3.8 and satisfies the Propositions 3.6 and 3.7, which serve as definitions within the statement of the theorem. We give our proof in the Appendix.

**Discussion.** We denote $\mathcal{E}_{tra} \subseteq \mathcal{E}_{avail}$ as the training set during the training process, and utilize $\mathcal{E}_{avail}$ as the target domain set for generalization. The non-generalized model obtained during this process is to minimize the variation between the outputs on $\mathcal{E}_{tra}$ and $\mathcal{E}_{avail}$, indicating variation across different domain sets.

## 5 GENERALIZATION INEQUALITY

In this section, we establish an architecture of the relationship between $\mathcal{E}_{tra}$ to $\mathcal{E}_{avail}$ through the generalization ratio. We first consider that our goal is to generalize $\mathcal{E}_{tra}$ to $\mathcal{E}_{avail}$, and analyze the classifier $f$ for minimizing generalization error defined by

$$err(f) = \max_{e \in \mathcal{E}_{avail}} \mathcal{L}(e, f) - \max_{e \in \mathcal{E}_{tra}} \mathcal{L}(e, f), \tag{7}$$

where if the loss function $\ell(\cdot, \cdot)$ is bounded by $[0, \mathcal{C}]$, we can get the upper bound for $err(f)$ in Ye et al. (2021). Before the inequality, we will demonstrate that despite differences in the distributions of the two ideal feature matrix functions, it is still possible to accurately predict the distribution of the output y in $\mathcal{E}_{avail}$, that is, regardless of variations in the distributions of the feature matrix functions, as long as the output y within $\mathcal{E}_{avail}$ can be accurately predicted, the models are considered generalized models defined in the Definition 3.3. We prove the following:

**Theorem 4.1** *(Variation on Distribution). Let Definition 3.3 hold. Suppose $W$ and $W'$ are ideal feature matrix functions of a model $f \in F$. If $\mathcal{V}_{KL}(W, \mathcal{E}_{avail}) = \mathcal{V}_{KL}(W', \mathcal{E}_{avail}) = 0$, then regardless of the distributions of the ideal feature matrix functions, even if $\mathbb{P}(W(X^{e''})) \neq \mathbb{P}(W'(X^{e''}))$ for all $e'' \in \mathcal{E}_{tra}$, the model is still a generalized model.*

The variations of the ideal feature matrix functions are zero, although the distributions may differ, the model can still output the distribution of $y$ which remains consistent in $\mathcal{E}_{avail}$. Thus, a generalized model still makes accurate predictions even in the presence of different distributions. Based on our formulation and previous work (Appendix 9 in Ye et al. (2021)), the following theorem demonstrates the upper bound for the loss on $\mathcal{E}_{avail}$, which is the set of domain we want to generalize to:

**Theorem 4.2** (Generalization Inequality). *Let Proposition3.6. and Theorem 4.1 hold. Suppose we have learned a base model $x' = h(x)$ and a classifier $f(x') = g_i(w_i(x'))$. Consider any loss satisfying $l(\hat{y}, y) = \sum_{k=1}^{K} l_0(\hat{y}_k, y_k)$ and is bounded by $[0, \mathcal{C}]$. For any linear top classifier $g_i \in \mathcal{G}$ decomposed from the non-generalized model $f \in F$, i.e.,*

$$f(x') = A_i w_i(x') + b_i, \tag{8}$$

*with $A_i \in \mathbb{R}^{K \times d}$, $b_i \in \mathbb{R}^K$, $w_i \in W$, where $A_i = \prod_{j=i+1}^{d} w_j$ is the product of all weight matrices, and $b_i = \sum_{j=i+1}^{d} \left( \prod_{k=1}^{j} w_k \right) \cdot b_j$ is bias vector, if $(\mathcal{E}_{tra}, \mathcal{E}_{avail})$ is $(GR_{KL}(W, \mathcal{E}_{avail}), \bar{\epsilon})$-learnable under $W$ with average variation and informativeness, then we can have*

$$\max_{e \in \mathcal{E}_{avail}} \mathcal{L}(e, f) \leq O\left( \mathcal{C} \cdot \frac{1}{d} \sum_{i=1}^{d} GR_{KL}(w_i, \mathcal{E}_{avail}) \right) + \max_{e \in \mathcal{E}_{tra}} \mathcal{L}(e, f), \tag{9}$$

*where $O(\cdot)$ is positive function and depends only on $d$.*

In calculating the upper bound for the loss on $\mathcal{E}_{avail}$, we use the generalization ratio combined with the linear top model theorem in Ye et al. (2021). This theorem calculates the average generalization ratio, and uses it to approximate the upper bound, with an additional term $\mathcal{C}$ corresponding to imbalance, thus increasing the generalization bound. The above theorem applies to all types of neural networks combined with any type of classifier (linear neural network). By calculating the upper bound of errors from both seen and unseen domains for model training, each training iteration takes into account the concept of generalization, thereby enabling the model to have generalization capabilities. The proof of Theorems 4.1 and 4.2 is in the Appendix.

**Discussion.** We reasonably connect the generalization ratio with $\mathcal{E}_{avail}$ using inequalities, intuitively expressing that the generalization problem is related to both variation and informativeness, with implications for the loss in unseen domains. Due to practical application limitations, Theorem 4.2 has certain constraints, as it only explains the behavior of the boundary in unseen domains under the assumptions of the corresponding domain. This may not fully reflect performance across all domains.

# 6 GENERALIZATION DECISION PROCESS (GDP)

In this section, we introduce the Generalization Decision Process (GDP), which incorporates the reinforcement learning method. In GDP, we set the generalization ratio as the environment, and select different losses from the training set $\mathcal{L}(e'', f), e'' \in \mathcal{E}_{tra}$ to minimize the loss from the available set $\mathcal{L}(\mathcal{E}_{avail}, f)$ based on the maximum cumulative reward.

We consider GDP that consists of $(\mathcal{G}, \mathcal{A}, U, r)$, where $\mathcal{G}$ is a generalization ratio space $\mathcal{G} \in \mathbb{R}^+$, and $\mathcal{A} = \{\mathcal{L}(e_i, f) | e_i \in \mathcal{E}_{tra}\}$ constitutes a finite action set of different losses from the training set, with the number of $e_i$ fixed. Specifically, we further let $G_t \in \mathcal{G}$ denote the state of generalization ratio at time $t$, and $A_t \in \mathcal{A}$ denote the action at time $t$. Let $U$ denote the action dependent transition function $U$ for the underlying process of generalization ratios $\{G_t\}_{t \geq 0}$, where $U : \mathcal{G} \times \mathcal{G} \times \mathcal{A} \times \mathcal{A} \to \mathcal{G}' \times \mathcal{A}'$, and $U$ outputs the pair $(g', a') \in \mathcal{G}' \times \mathcal{A}'$, i.e., $U((g_{t-1:t}), (a_{t-1:t})) = (g'_t, a'_t)$. Finally, we define the one-stage reward at time $t$ as $\mathcal{R}(G'_t, A'_t)$, where the reward function $\mathcal{R} : \mathcal{G}' \times \mathcal{A}' \to \mathbb{R}$ is defined to be uniformly bounded, i.e., $\forall (g', a') \in \mathcal{G}' \times \mathcal{A}'$, we have $\mathcal{R}(g', a') \in [-r_{max}, r_{max}]$.

Then, we define a stationary policy that maps a state $g \in \mathcal{G}$ to a probability distribution $\pi(\cdot | g)$ over $A$, which does not depend on time. Given a policy $\pi$, we define the corresponding value function

$V_\pi : \mathbb{R} \to \mathbb{R}$ as the expected total reward from before obtained by actions executed according to $\pi : V^\pi(g_0) = \mathbb{E}_\pi \left[ \sum_{i=0}^t \mathcal{R}(g_i', a_i') \big| G_0 = g_0 \right]$. We also define action-value function $\mathcal{Q}^\pi : \mathcal{G}' \times \mathcal{A}' \to \mathbb{R}$ as $\mathcal{Q}^\pi(g_t', a_t') = V_\pi(g_0) + \mathcal{R}(U(g_t, g_{t-1}a_t, a_{t-1}))$. Our goal is to use maximum the value from the value function to find an optimal policy that minimizes the loss on available set $\mathcal{L}(\mathcal{E}_{avail}, f)$ for backpropagation. The optimal policy $\pi^*$ is then greedy with respect to $\mathcal{Q}^*$.

To combine the concept of transitions on generalized states with $U$, we use the generalized ratio and $\mathcal{A}$ across preceding and subsequent time steps to get their respective changes, i.e., $U$ is also a real-valued function defined on $\mathcal{G} \times \mathcal{G} \times \mathcal{A} \times \mathcal{A}$, which outputs the change in generalization ratio $G'$ and the change in training loss $A'$ between different iterations, i.e., $\forall t \in T, t \geq 1, \exists (G_t, G_{t-1}, A_t, A_{t-1}) \in \mathcal{G} \times \mathcal{G} \times \mathcal{A} \times \mathcal{A}$, we have $U(G_t, G_{t-1}, A_t, A_{t-1}) = ((G_t - G_{t-1}), (A_t - A_{t-1})) = (G', A') \in \mathcal{G}' \times \mathcal{A}'$. Then, we input $(G', A')$ into the reward function $R$ and introduce Theorem 4.2 to get the change in $\mathcal{L}(\mathcal{E}_{avail}, f)$. Based on the change in $\mathcal{L}(\mathcal{E}_{avail}, f)$ is positive or negative, we define "generalized transition" and "non-generalized transition":

**Definition 5.1** (Generalization Transition). *The transition from $(G_{t-1}, A_{t-1})$ to $(G_t, A_t)$ is a generalization transition if there exists a $\Delta \max_{e \in \mathcal{E}_{tra}} \mathcal{L}(e, f)$, we have a generalization inequality such that $\Delta O(GR_{KL}(W, \mathcal{E}_{avail})) < 0$.*

**Definition 5.2** (Non-generalization Transition). *The transition from $(G_{t-1}, A_{t-1})$ to $(G_t, A_t)$ is a non-generalization transition if there exists a $\Delta \max_{e \in \mathcal{E}_{tra}} \mathcal{L}(e, f)$, we have a generalization inequality such that $\Delta O(GR_{KL}(W, \mathcal{E}_{avail})) > 0$. If $\Delta \max_{e \in \mathcal{E}_{tra}} \mathcal{L}(e, f) > 0$, we call it an underfitting transition. If $\Delta \max_{e \in \mathcal{E}_{tra}} \mathcal{L}(e, f) < 0$, we also call it the overfitting transition.*

We assign positive rewards to generalized transitions and negative rewards to non-generalized transitions, allowing them to effectively integrate within the action-value function space. The reward function $R$ is defined in $\mathcal{G}' \times \mathcal{A}'$, which inputs the changes in the generalization ratio and provides different rewards based on whether the transitions are positive or negative. Therefore, we let the lowest $\mathcal{L}(e'', f)$ correspond to the highest positive reward in a generalization transition. Conversely, we let the highest $\mathcal{L}(e'', f)$ correspond to the highest negative reward in a non-generalization transition.

In the generalization transition, we minimize $\mathcal{L}(e'', f)$ to allow the model to achieve a good generalization performance in $\mathcal{L}(\mathcal{E}_{avail}, f)$. However, in the non-generalization transition, we maximize $\mathcal{L}(e'', f)$ to negate this round of backpropagation, thereby regaining the generalization transition and allowing subsequent backpropagation to proceed. Although minimizing $\mathcal{L}(e'', f)$ during the non-generalization transition may also reduce the loss on $\mathcal{E}_{avail}$, the primary focus during backpropagation is on reducing the loss on $\mathcal{E}_{tra}$. Therefore, the final model after backpropagation is unable to achieve good generalization ability.

The probability distribution $\pi(\cdot|g)$ of the stationary policy assigns equal probabilities to every action $A_t \in \mathcal{A}$. If we only use $a^* = \max_a \mathcal{Q}^\pi(g, a)$ as the criterion, it removes the impact of the chance on other actions. To address this, we introduce the $\epsilon$-greedy policy to replace it. After the process of action-value function, for each state $G_t \in \mathcal{G}$, we select the best action based on the action-value function with the probability of $1 - \epsilon$, i.e.,

$$\pi(a|g) = \begin{cases} 1 - \epsilon + \frac{\epsilon}{\|A\|}, & \text{if } a^* = \arg\max_{a \in A} \mathcal{Q}^\pi(g, a) \\ \frac{\epsilon}{\|A\|}, & \text{otherwise} \end{cases} \tag{10}$$

where $\epsilon \in [0, 1]$ is a hyperparameter, and $\|A\| = k \geq 2$ denotes the number of actions. $\epsilon$ is the exploration factor that controls the probability of selecting a random action. $\arg\max_{a \in A} \mathcal{Q}^\pi(g, a)$ represents the action $a^*$ that maximizes the action-value function $\mathcal{Q}^\pi(g, a)$ in the state $g$.

## 7 GDP FOR BACKPROPAGATION WITH GRADIENT DESCENT

We introduce a generalization gradient descent (GGD) algorithm (see Algorithm 1), which is a framework on GDP for backpropagation and uses gradient descent to train the model. GGD can use $\mathcal{L}(e'', f)$ for backpropagation to find the gradient and optimize each weight using gradient descent. This not only considers the loss itself on $\mathcal{E}_{tra}$ during model training, but also takes into account the loss generalized to $\mathcal{E}_{avail}$. In addition, Algorithm 1 for training the model, we calculate its generalization ratio and $\mathcal{L}(e'', f)$. We then incorporate them into GDP. Using the $\epsilon$-greedy strategy,

---

**Algorithm 1:** Generalization Gradient Descent (GGD)

---

**Input:** Small $\bar{\epsilon} > 0$, base model $h$, classifier $f_W$, observations $\bar{e}' = \left\{ \left( h(X^{e_i'}), Y^{e_i'} \right) \right\}_{i=1}^{n'}$ from unseen domains $\bar{e}' \subseteq \mathcal{E}_{val}$, observations $\bar{e}'' = \left\{ \left( h(X^{e_i''}), Y^{e_i''} \right) \right\}_{i=1}^{n''}$ from seen domains $\bar{e}'' \subseteq \mathcal{E}_{tra}$, epoches $E$, batch_size $b$, training iterations $T = n''E/b$, number of actions $k$ and available set $\mathcal{E}_{avail} = \mathcal{E}_{val} \dot{\cup} \mathcal{E}_{tra}$ ( $n'' > n' \geq 1$ are positive integers).

**Initialization**: $\forall G_t \in \mathcal{G}, \forall a_i \in \mathcal{A}, \{e_j''\}_{j=i}^{i+k-1} \subseteq \bar{e}''$ and $e_j'' \neq e_{j'}''$ for all $j \neq j'$

$\pi \leftarrow$ an $\epsilon$-greedy policy

$A_0 \leftarrow$ Randomly select $\mathcal{L}(e'', f)$, where $e'' \in \bar{e}''$

$G_0 \leftarrow 0$

**for** *time $t$ from* $1$ **to** $T$ **do**

    {# batch_size = 1}

    **for** *number $i$ from* $1$ **to** $(n'' - k + 1)$ **do**

        $G_t \leftarrow GR_{KL}(W, \mathcal{E}_{avail})$ {# Definition 3.8 (Generalization Ratio)}

        **for** *number $j$ from $i$* **to** $(i + k - 1)$ **do**

            Compute empirical risk: $a_j \leftarrow \mathcal{L}\left(e_j'', f\right)$

        **end**

        $Q(G_t, a_i) \leftarrow Q(G_{t-1}, A_{t-1}) + \mathcal{R}(U(G_{t-1}, G_t A_{t-1}, a_j))$

        $A^* \leftarrow argmax_{a_j \in \mathcal{A}} Q(G_t, a_j)$

        **For all** $a_i \in \mathcal{A}$:

        $\pi(a_j | G_t) \leftarrow \begin{cases} 1 - \epsilon + \epsilon / \|A\|, & if \ a_j = A^* \\ \epsilon / \|A\|, & if \ a_j \neq A^* \end{cases}$

        $A_t \leftarrow \pi(a_j | G_t)$

        Backpropagate gradients $\nabla_W A_t$ in the classifier $f$ with standard PyTorch

    **end**

**end**

---

we select the optimal loss to adjust the weights and obtain the best model, with $T$ as the fixed number of iterations and $b$ as the batch size.

# 8 EXPERIMENT

In our experiment, we demonstrate the Generalization Gradient Descent (GGD) algorithm on Colored MNIST (Arjovsky et al., 2019) and CIFAR10 (Ho-Phuoc, 2018) datasets, and use two neural networks, LLNet and LANet. The LLNet model consists of 7 linear layers acting as the classifier. The first layer flattens the input image of size $3{\times}28{\times}28$ for Colored MNIST ($3{\times}32{\times}32$ for CIFAR10) and outputs 1000 neurons. The subsequent layers respectively produce outputs of 500, 100, 50, 25, 20, and 10 neurons. The LANet model is composed of a base model and a classifier. The base model consists 5 linear layers and 4 ReLU activation layers, while the classifier consists of 2 linear layers. The first layer flattens the input image of size $3{\times}28{\times}28$ ($3{\times}32{\times}32$ for CIFAR10) and outputs 1000 neurons. The subsequent layers respectively produce outputs of 500, 100, 50, 25, 20, and 10 neurons.

**Settings** We use the GGD algorithm and the traditional gradient descent (TGD) algorithm to train our models. The function $\mathcal{R}(g', a')$ operates within $[-1, 1]$ with $k = 2$ actions. We employ CrossEntropyLoss and Adam optimizer with learning rate = 0.001 and batch_ size = 5. In addition, we use the experimental form of features (Theorem 2.1) to calculate the generalization ratio. Our experiments are conducted on the Google Colab platform using an L4 GPU, aim to evaluate the effectiveness of the GGD algorithm in these models.

**Result** We discuss the experiments and results in in-distribution and out-of-distribution experiments. In our in-distribution experiments on both Colored MNIST and CIFAR10, we divide each dataset into training, validation, and test sets, ensuring that all sets are i.i.d. in the experiments. We train our models over 5 epochs, with the results shown in Table 1, 2. The experiments demonstrate that GGD has in-distribution generalization capabilities. On the other hand, in our out-of-distribution experiments, we mainly used the Colored MNIST task to predict whether a digit is correct. Crucially, the color of the digit is spuriously correlated with the label: the correlation strength varies between the two training domains $\mathcal{E} = \{90\%, 80\%\}$. To test whether the model learned to ignore the color,

Table 1: In-Distribution Experiments Results. We present the best results achieved during training, and the results "(R)" after final training, when overfitting occurs. "C_MNIST" denotes the Colored MNIST dataset.

| Model | Accuracy | C_MNIST | CIFAR10 | C_MNIST (R) | CIFAR10 (R) |
|---|---|---|---|---|---|
| LLNet (GGD) | Train acc. | **90.44%** | 42.25% | **86.72%** | **40.65%** |
| LLNet (GGD) | Test acc. | 87.37% | 39.58% | **84.31%** | **38.74%** |
| LLNet (TGD) | Train acc. | 90.33% | **42.41%** | 76.33% | 39.68% |
| LLNet (TGD) | Test acc. | **87.99%** | **39.79%** | 73.45% | 37.43% |
| LANet (GGD) | Train acc. | **93.61%** | **74.51%** | **92.48%** | **73.59%** |
| LANet (GGD) | Test acc. | **91.58%** | **54.47%** | **90.34%** | **52.89%** |
| LANet (TGD) | Train acc. | 91.02% | 58.79% | 91.32% | 56.69% |
| LANet (TGD) | Test acc. | 89.89% | 51.98% | 88.91% | 50.10% |

Table 2: In-Distribution Experiments Results. "(avg)" denotes the average of the best results achieved during training and the results after final training, when overfitting occurs. "inc" represents the improvement "↑" brought by the GGD algorithm.

| Model | Metric | C_MNIST (avg) | acc. inc | CIFAR10 (avg) | acc. inc |
|---|---|---|---|---|---|
| LLNet (GGD) | Train acc. | **88.58%** | 5.25%↑ | **41.45%** | 0.41%↑ |
| LLNet (GGD) | Test acc. | **85.84%** | 5.12%↑ | **39.16%** | 0.55%↑ |
| LLNet (TGD) | Train acc. | 83.33% | - | 41.04% | - |
| LLNet (TGD) | Test acc. | 80.72% | - | 38.61% | - |
| LANet (GGD) | Train acc. | **93.04%** | 1.87%↑ | **74.05%** | 16.31%↑ |
| LANet (GGD) | Test acc. | **90.96%** | 1.69%↑ | **53.68%** | 2.64%↑ |
| LANet (TGD) | Train acc. | 91.17% | - | 57.74% | - |
| LANet (TGD) | Test acc. | 89.40% | - | 51.04% | - |

Table 3: Out-of-Distribution Experiments Results. We present the best results achieved during training. "$\mathcal{E}_r$" denotes the experiment where red is used as the test set, while blue and green are used as the training set. "(80%)" indicates the correlation Propositionwithin the training set. "LL" refers to the LLNet, and "LA" refers to the LANet.

| Model | Accuracy | $\mathcal{E}_r(80\%)$ | $\mathcal{E}_r(90\%)$ | $\mathcal{E}_g(80\%)$ | $\mathcal{E}_g(90\%)$ | $\mathcal{E}_b(80\%)$ | $\mathcal{E}_b(90\%)$ |
|---|---|---|---|---|---|---|---|
| LL(GGD) | Train acc. | **93.73%** | 95.33% | **93.81%** | **94.66%** | **90.49%** | 94.71% |
| LL(GGD) | Test acc. | **61.86%** | **69.48%** | **63.43%** | **68.04%** | **68.23%** | **68.26%** |
| LL(TGD) | Train acc. | 93.59% | **95.48%** | 93.30% | 93.73% | 90.27% | **95.10%** |
| LL(TGD) | Test acc. | 55.07% | 67.08% | 62.62% | 55.35% | 62.42% | 67.28% |
| LA(GGD) | Train acc. | 93.99% | 94.99% | 94.11% | 94.54% | 90.57% | **95.19%** |
| LA(GGD) | Test acc. | **59.91%** | **60.65%** | **63.01%** | **67.18%** | **64.97%** | **66.01%** |
| LA(TGD) | Train acc. | **94.96%** | **95.81%** | **94.57%** | 94.42% | **91.54%** | 94.00% |
| LA(TGD) | Test acc. | 56.47% | 56.20% | 58.37% | 60.59% | 59.87% | 59.44% |

we reversed this correlation during the testing. In brief, a model biased towards considering only the color achieves a test accuracy of 10%, whereas an oracle model that perfectly predicts the shape achieves a test accuracy of 75%. Furthermore, we tested with blue, green, and red as test sets (unseen domains), maintaining a correlation 80% or 90% between the digit's color and the label. We train our models over 10 epochs, with the results summarized in Table 3, 4 the dynamics shown in Figure 1. The experiments demonstrate that GGD has out-of-distribution generalization capabilities and that the Generalization Ratio can serve as an indicator of generalization. Our algorithm effectively selects the optimal loss function in both in-distribution and out-of-distribution experiments, resulting in the best testing accuracy. In addition, sources of variability, such as the epoch, the reward size, and the amount of data in the validation set, can also lead to different results.

**Limitations and Future Works** In our formulation, we used the conditional probabilities of features to address the generalization problem, with less focus on discussing the relationship between model

Table 4: Out-of-Distribution Experiments Results. "(avg*)" denotes the average of the correlation assumptions (80%) and (90%) in the training set. "avg*" represents the average of the experiments where red, green, and blue are used as the test set.

| Model | Accuracy | $\mathcal{E}_r$(avg*) | $\mathcal{E}_g$(avg*) | $\mathcal{E}_b$(avg*) | avg* | acc. inc |
|---|---|---|---|---|---|---|
| LL(GGD) | Train acc. | 94.53% | **94.23%** | 92.60% | **93.78%** | 0.21% ↑ |
| LL(GGD) | Test acc. | **65.67%** | **65.73%** | 68.24% | **66.54%** | 4.91% ↑ |
| LL(TGD) | Train acc. | 94.53% | 93.51% | **92.68%** | 93.57% | - |
| LL(TGD) | Test acc. | 61.07% | 58.98% | 64.85% | 61.63% | - |
| LA(GGD) | Train acc. | 94.49% | 94.32% | **92.88%** | 93.89% | 0.32% ↓ |
| LA(GGD) | Test acc. | 60.28% | **65.09%** | 65.49% | 63.62% | 5.14% ↑ |
| LA(TGD) | Train acc. | **95.38%** | **94.49%** | 92.77% | **94.21%** | - |
| LA(TGD) | Test acc. | 56.33% | 59.48% | 59.65% | 58.48% | - |

parameters and the generalization state of the classifier model to obtain precise information about the overall features. The proposed GDP might lead to uncertainty in the generalization results of the model due to varying rewards. We hope to address these issues in future research.

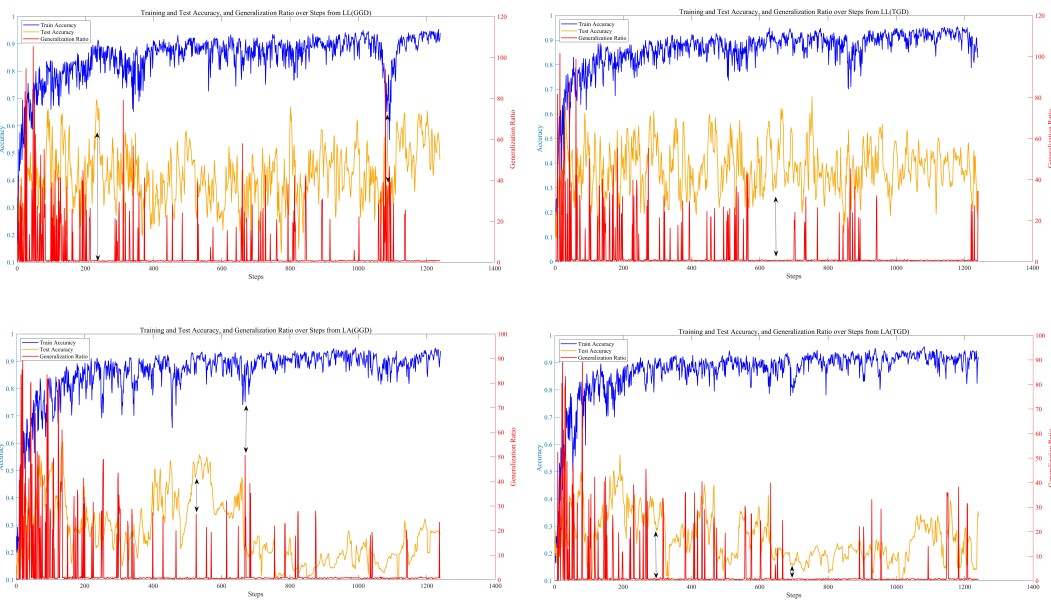

Figure 1: Training Dynamics. We observed the model training dynamics on the $\mathcal{E} = \{90\%\}$ Colored MNIST dataset. In the GGD algorithm, the Generalization Ratio (red) clearly highlights steps where the training accuracy (blue) and test accuracy (orange) are low, indicating improved performance (double arrow). In contrast, the TGD algorithm, which does not use this metric, showed no such correlation (double arrow). The GGD algorithm outperforms TGD in both test accuracy and generalization ratio, demonstrating more consistent performance and fewer fluctuations.

# 9 CONCLUSION

We take the first step towards combining generalization with reinforcement learning to train models, and propose a generalization ratio to quantitatively characterize the degree of generalization and a Generalization Decision Process to formalize the relationship between loss in seen and unseen domains. Based on our framework, we derive a generalization inequality and design a Generalization Gradient Descent algorithm to optimize loss selection for backpropagation. Finally, the results of our experiments show that our algorithm significantly outperforms traditional gradient descent methods.

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

# A  APPENDIX

In this section, we provide complete proofs of our assumptions and theorems.

## A.1  PROOF OF THEOREM 2.1

**Theorem 2.1** (Experimental Form of Features). *Let $e_1, \ldots, e_k \in \mathcal{E}$ be domains such that $P(e_i) > 0$, $i = 1, 2, \ldots, k$. Assume further that $e_1, \ldots, e_k$ also form a partition of the domain $\mathcal{E}$, and $\forall e_i, e_j \in \mathcal{E}$ where $e_i \neq e_j$, they are distinct domains. Let $y$ be any class in $\mathcal{Y}$ and $W$ be a feature matrix function. Then we have the conditional probability of features given a class as follows:*

$$\mathbb{P}(W(X^{e_i})|y) = \frac{P(W(X^{e_i}))P(y|W(X^{e_i}))}{\sum_{i=1}^{k}(P(W(X^{e_i}))P(y|W(X^{e_i})))}. \tag{11}$$

*proof:* Given that $W$ contains many linear feature extractors, we assume $W$ is a one-to-one function, ie $\forall X^{e_i}, X^{e_j} \in \mathcal{X}$, $W(X^{e_i}) \neq W(X^{e_j})$. Base on the Propositionthat for all $e_i, e_j \in \mathcal{E}$ where $e_i \neq e_j$, we have $X^{e_i} \neq X^{e_j}$, such that $\mathbb{P}(W(X^{e_i})|y) \geq 0$, $\mathbb{P}(y|y) = 1$, and $\mathbb{P}\left(\bigcup_{i=1}^{k} W(X^{e_i}) \Big| y\right) = \sum_{i=1}^{k} \mathbb{P}(W(X^{e_i})|y)$. Using the definition of conditional probability, we have

$$\mathbb{P}(W(X^{e_i})|y) = \frac{P(W(X^{e_i})\bigcap y)}{P(y)} \tag{12}$$

Hence, by the law of total probability, the result follows

$$\mathbb{P}(W(X^{e_i})|y) = \frac{P(W(X^{e_i})\bigcap y)}{P(y)}$$
$$= \frac{P(W(X^{e_i}))P(y \mid W(X^{e_i}))}{\sum_{i=1}^{k}\left(P(W(X^{e_i}))P(y \mid W(X^{e_i})) + P(W(X^{e_i'}))P(y \mid W(X^{e_i'}))\right)} \tag{13}$$

The proof is finished. ∎

## A.2 PROOF OF PROPOSITION3.6

**Proposition3.6.** *Suppose $\mathcal{E}_{val} \neq \phi$ and $\mathcal{E}_{tra} \neq \phi$, with $\mathcal{E}_{val}$ and $\mathcal{E}_{val}$ are not i.i.d. If $\mathcal{E}_{val}$ is the domain set we want to generalize to, such that $\forall e' \in \mathcal{E}_{val}$, $\forall e'' \in \mathcal{E}_{tra}$, and $\exists y \in \mathcal{Y}$, where $p\left(y\middle|W\left(X^{e'}\right)\right) \neq p\left(y\middle|W\left(X^{e''}\right)\right)$, then $KL\left(\mathbb{P}\left(W\left(X^{e'}\right)\middle|y\right), \mathbb{P}\left(W\left(X^{e''}\right)\middle|y\right)\right) \neq 0$.*

*Proof.* Base on the definition that $KL(P\|Q) = \sum_x P(x)\log\left(\frac{P(x)}{Q(x)}\right) = 0 \Leftrightarrow P = Q$. If there exists $y \in \mathcal{Y}$, such that for all $e' \in \mathcal{E}_{val}$, and $e'' \in \mathcal{E}_{tra}$, $P\left(y\middle|W\left(X^{e'}\right)\right) \neq P\left(y\middle|W\left(X^{e''}\right)\right)$, we have $KL\left(\mathbb{P}\left(W\left(X^{e'}\right)\middle|y\right), \mathbb{P}\left(W\left(X^{e''}\right)\middle|y\right)\right) \neq 0$ Hence, for any $P\left(y\middle|W\left(X^{e'}\right)\right) \neq P\left(y\middle|W\left(X^{e''}\right)\right)$, $KL\left(\mathbb{P}\left(W\left(X^{e'}\right)\middle|y\right), \mathbb{P}\left(W\left(X^{e''}\right)\middle|y\right)\right) \neq 0$. The proof is finished. ∎

## A.3 PROOF OF PROPOSITION3.7

**Proposition3.7.** *Suppose the conditions of $\mathcal{E}_{val}$ and $\mathcal{E}_{tra}$ in Proposition3.6. hold. If we can get a generalized model $f \in F$ such that $\forall e' \in \mathcal{E}_{val}$, $\forall e'' \in \mathcal{E}_{tra}$, and $\forall y \in \mathcal{Y}$, we have $p\left(y\middle|W\left(X^{e'}\right)\right) = p\left(y\middle|W\left(X^{e''}\right)\right)$, then it follows that $KL\left(\mathbb{P}\left(W\left(X^{e'}\right)\middle|y\right), \mathbb{P}\left(W\left(X^{e''}\right)\middle|y\right)\right) = 0$.*

*Proof.* Base on the definition that $KL(P\|Q) = \sum_x P(x)\log\left(\frac{P(x)}{Q(x)}\right) = 0 \Leftrightarrow P = Q$. If there exists $y \in \mathcal{Y}$, such that for all $e' \in \mathcal{E}_{val}$, and $e'' \in \mathcal{E}_{tra}$, $P\left(y\middle|W\left(X^{e'}\right)\right) = P\left(y\middle|W\left(X^{e''}\right)\right)$, we have $KL\left(\mathbb{P}\left(W\left(X^{e'}\right)\middle|y\right), \mathbb{P}\left(W\left(X^{e''}\right)\middle|y\right)\right) = 0$ Hence, for any $P\left(y\middle|W\left(X^{e'}\right)\right) \neq P\left(y\middle|W\left(X^{e''}\right)\right)$, $KL\left(\mathbb{P}\left(W\left(X^{e'}\right)\middle|y\right), \mathbb{P}\left(W\left(X^{e''}\right)\middle|y\right)\right) = 0$. The proof is finished. ∎

## A.4 PROOF OF THEOREM 3.9

**Theorem 3.9** (Expansion Function). *The generalization ratio is an expansion function (Definition 3.4 in Ye et al. (2021)).*

*Proof.* In the following, we show that $GR_{KL}$ satisfies the mathematical and functional properties of an expansion function.

First, $GR_{KL} : \mathbb{R}^+ \cup \{0\} \to \mathbb{R}^+ \cup \{0, +\infty\}$ is a function with the following properties: 1) $GR_{KL}(\cdot)$ is monotonically increasing and $GR_{KL}(\cdot) \geq \mathcal{V}(\cdot), \forall \mathcal{V}(\cdot) \geq 0$; 2) $\lim_{\mathcal{V}(\cdot) \to 0^+} GR_{KL}(\cdot) = 0$. Hence, $GR_{KL}$ is mathematically an expansion function.

Next, we know that the variation and informativeness in the domain set $\mathcal{E}_{all} \supseteq \mathcal{E}_{avail}$ are defined as follows:

Definition (Variation). *The variation of the set of scalar maps $W(\cdot)$ from the available domain set $\mathcal{E}_{avail}$ to the all domain set set $\mathcal{E}_{all}$ is*

$$\mathcal{V}_{KL}(W, \mathcal{E}_{all}) = \max_{y \in \mathcal{Y}} \sup_{\substack{e' \in \mathcal{E}_{all} \setminus \mathcal{E}_{avail}, \\ e'' \in \mathcal{E}_{avail}}} KL\left(\mathbb{P}\left(W\left(X^{e'}\right)\middle|y\right), \mathbb{P}\left(W\left(X^{e''}\right)\middle|y\right)\right), \quad (14)$$

*where KL is KL-divergence and we concern the set relation rather than the method used for splitting.*

Definition (Informativeness). *The informativeness of the set of scalar maps $W(\cdot)$ across the available domain set $\mathcal{E}_{avail}$ is*

$$\mathcal{I}_{KL}(W, \mathcal{E}_{avail}) = \frac{1}{n(n-1)}\left(\sum_{\substack{y \neq y' \\ y,y' \in \mathcal{Y}}} \min_{e'' \in \mathcal{E}_{avail}} KL\left(\mathbb{P}(W(X^{e''})\mid y), \mathbb{P}(W(X^{e''})\mid y')\right)\right) + \bar{\epsilon},$$

$$(15)$$

*where $\bar{\epsilon} > 0$ is approximately equal to zero positive value, and KL is KL-divergence.*

Then, if we have a non-generalized model $f \in F$, for any $\mathcal{E}_{tra} \subseteq \mathcal{E}_{avail} \subseteq \mathcal{E}_{all}$, we have

$$\mathcal{V}_{KL}(W, \mathcal{E}_{avail}) \leq \mathcal{V}_{KL}(W, \mathcal{E}_{all}) \quad (16)$$

and

$$\mathcal{I}_{KL}(W, \mathcal{E}_{tra}) \geq \mathcal{I}_{KL}(W, \mathcal{E}_{avail}) \quad (17)$$

at initialization. Thus, we can obtain

$$\frac{\mathcal{V}_{KL}\left(W, \mathcal{E}_{avail}\right)}{\mathcal{I}_{KL}\left(W, \mathcal{E}_{tra}\right)} \leq \frac{\mathcal{V}_{KL}\left(W, \mathcal{E}_{all}\right)}{\mathcal{I}_{KL}\left(W, \mathcal{E}_{avail}\right)}. \tag{18}$$

Plugging $\mathcal{V}_{KL}\left(W, \mathcal{E}_{avail}\right)$ and $\mathcal{V}_{KL}\left(W, \mathcal{E}_{all}\right)$ into both sides, we can rewrite the inequality as

$$\mathcal{V}_{KL}\left(W, \mathcal{E}_{avail}\right) + \frac{\mathcal{V}_{KL}\left(W, \mathcal{E}_{avail}\right)}{\mathcal{I}_{KL}\left(W, \mathcal{E}_{tra}\right)} \leq \mathcal{V}_{KL}\left(W, \mathcal{E}_{all}\right) + \frac{\mathcal{V}_{KL}\left(W, \mathcal{E}_{all}\right)}{\mathcal{I}_{KL}\left(W, \mathcal{E}_{avail}\right)}. \tag{19}$$

Since we aim to generalize to the entire domain set $\mathcal{E}_{all}$, and our final goal is $\mathcal{V}_{KL}\left(W, \mathcal{E}_{all}\right) \to 0$ and $\mathcal{I}_{KL}\left(W, \mathcal{E}_{avail}\right) \to +\infty$ in $\mathcal{E}_{all}$, we have

$$\mathcal{V}_{KL}\left(W, \mathcal{E}_{avail}\right) + \frac{\mathcal{V}_{KL}\left(W, \mathcal{E}_{avail}\right)}{\mathcal{I}_{KL}\left(W, \mathcal{E}_{tra}\right)} \leq \mathcal{V}_{KL}\left(W, \mathcal{E}_{all}\right). \tag{20}$$

Finally, we also observe that $\mathcal{V}_{KL}\left(W, \mathcal{E}_{avail}\right) \to 0$ and $\mathcal{I}_{KL}\left(W, \mathcal{E}_{tra}\right) \to +\infty$ during the training process, so we have

$$\mathcal{V}_{KL}\left(W, \mathcal{E}_{avail}\right) \leq \mathcal{V}_{KL}\left(W, \mathcal{E}_{all}\right). \tag{21}$$

Given the assumed properties of the expansion function, the generalization of OOD can be guaranteed because we can predict whether an invariant and informative feature in $\mathcal{E}_{avail}$ will vary significantly in the unseen domain $\mathcal{E}_{all}$. The proof is finished. ∎

## A.5 Proof of Theorem 3.10

**Theorem 3.10** (Learnability on Non-generalized Model). *Let $W$ be a feature matrix function of the non-generalized model $f \in F$ defined in 3.2. Assuming $\mathcal{E}_{avail}$ to be the available domain set and $\mathcal{E}_{tra} = \mathcal{E}_{avail} \backslash \mathcal{E}_{val}$ to be the training set, we say that the non-generalized model is learnable if there exists a generalization ratio $GR_{KL}\left(W, \mathcal{E}_{avail}\right)$ and $\bar{\epsilon} > 0$, such that: given $W$ satisfy $\mathcal{I}_{KL}\left(W, \mathcal{E}_{tra}\right) \geq \bar{\epsilon}$, we have $GR_{KL}\left(W, \mathcal{E}_{avail}\right) > \mathcal{V}_{KL}\left(W, \mathcal{E}_{avail}\right)$. If such the generalization ratio $GR_{KL}\left(W, \mathcal{E}_{avail}\right)$ and $\bar{\epsilon}$ exist, we also call this model $(GR_{KL}\left(W, \mathcal{E}_{avail}\right), \bar{\epsilon})$-learnable.*

*Proof.* Given the generalization ratio $GR_{KL}\left(W, \mathcal{E}_{avail}\right)$ and $\bar{\epsilon} > 0$, we can derive that the generalization ratio $GR_{KL}\left(W, \mathcal{E}_{avail}\right)$ is an expansion function through the properties in the definition of expansion function (Definition 3.3 in Ye et al. (2021)). Then, $W$ is the feature matrix function containing all linear feature extractors from the model $f \in F$, and $\mathcal{I}_{KL}\left(W, \mathcal{E}_{tra}\right) \geq \bar{\epsilon}$. Thus, we have $GR_{KL}\left(W, \mathcal{E}_{avail}\right) > \mathcal{V}_{KL}\left(W, \mathcal{E}_{avail}\right)$ by the definition of learnability (Definition 3.4 in Ye et al. (2021)). Hence, this model is $GR_{KL}(W, \mathcal{E}_{avail}), \bar{\epsilon})$-learnable. The proof is finished. ∎

## A.6 Proof of Theorem 4.1

**Theorem 4.1** (Variation on Distribution). *Let Definition 3.3 hold. Suppose $W$ and $W'$ are ideal feature matrix functions of a model $f \in F$. If $\mathcal{V}_{KL}\left(W, \mathcal{E}_{avail}\right) = \mathcal{V}_{KL}\left(W', \mathcal{E}_{avail}\right) = 0$, then regardless of the distributions of the ideal feature matrix functions, even if $\mathbb{P}(W(X^{e''})) \neq \mathbb{P}(W'(X^{e''}))$ for all $e'' \in \mathcal{E}_{tra}$, the model is still a generalized model.*

*Proof.* Base on the Definition 3.3, we assume a generalized model $f \in F$. Given two different ideal feature matrix functions $W, W' \in \mathcal{W}^*$, $\forall e' \in \mathcal{E}_{val}$, $\forall e'' \in \mathcal{E}_{tra}$, $\forall y \in \mathcal{Y}$, we can obtain

$$P\left(y \middle| W\left(X^{e'}\right)\right) = P\left(y \middle| W'\left(X^{e'}\right)\right) = P\left(y \middle| W\left(X^{e''}\right)\right) \tag{22}$$

and

$$P\left(y \middle| W'\left(X^{e'}\right)\right) = P\left(y \middle| W\left(X^{e'}\right)\right) = P\left(y \middle| W'\left(X^{e''}\right)\right). \tag{23}$$

Therefore, we have

$$P\left(y \middle| W\left(X^{e''}\right)\right) = P\left(y \middle| W'\left(X^{e''}\right)\right) \tag{24}$$

By the definition of variation (Definition 3.4) and Proposition 3.7, the model f is $\mathcal{V}_{KL}\left(W, \mathcal{E}_{avail}\right) = 0$, but $W \neq W'$. Hence, $W$ causes the distribution to vary a lot in $\mathcal{E}_{tra}$ or not, the model is still a generalized model. ∎

## A.7 Proof of Theorem 4.2

**Theorem 4.2** (Generalization Inequality). *Let Proposition3.6. and Theorem 4.1 hold. Suppose we have learned a base model $x' = h(x)$ and a classifier $f(x') = g_i(w_i(x'))$. Consider any loss satisfying $l(\hat{y}, y) = \sum_{k=1}^{K} l_0(\hat{y}_k, y_k)$ and is bounded by $[0, \mathcal{C}]$. For any linear top classifier $g_i \in \mathcal{G}$ decomposed from the non-generalized model $f \in F$, i.e.,*

$$f(x') = A_i w_i(x') + b_i, \tag{25}$$

*with $A_i \in \mathbb{R}^{K \times d}$, $b_i \in \mathbb{R}^K$, $w_i \in W$, where $A_i = \prod_{j=i+1}^{d} w_j$ is the product of all weight matrices, and $b_i = \sum_{j=i+1}^{d} \left( \prod_{k=1}^{j} w_k \right) \cdot b_j$ is bias vector, if $(\mathcal{E}_{tra}, \mathcal{E}_{avail})$ is $(GR_{KL}(W, \mathcal{E}_{avail}), \bar{\epsilon})$-learnable under $W$ with average variation and informativeness, then we can have*

$$\max_{e \in \mathcal{E}_{avail}} \mathcal{L}(e, f) \leq O\left( \mathcal{C} \cdot \frac{1}{d} \sum_{i=1}^{d} GR_{KL}(w_i, \mathcal{E}_{avail}) \right) + \max_{e \in \mathcal{E}_{tra}} \mathcal{L}(e, f), \tag{26}$$

*where $O(\cdot)$ is positive function and depends only on $d$.*

*Proof.* Suppose we have learned a base model $x' = h(x)$ .Given a non-generalized model $f$, it can be decomposed similarly to $g \circ w$. Combining all kind of $g_i \circ w_i$, $i = 1, \ldots, d$ into one formula to represent $f(x')$ we get

$$f(x') = A_i w_i(x') + b_i \tag{27}$$

with $A_i \in \mathbb{R}^{K \times d}$, $b_i \in \mathbb{R}^K$, $w_i \in W$, where $A_i = \prod_{j=i+1}^{d} w_j$ is the product of all weight matrices, and $b_i = \sum_{j=i+1}^{d} \left( \prod_{k=1}^{j} w_k \right) \cdot b_j$ is the bias vector. Based on Theorem 4.1 and the linear top model theorem presented in Ye et al. (2021), as well as the learnability results for the non-generalized model (Theorem 3.1), for any kind of $g_i \circ w_i$, where $g_i$ is the top linear classifier, and $w_i$ is the linear feature extractor, we have the following inequality:

$$\max_{e \in \mathcal{E}_{avail}} \mathcal{L}(e, f) \leq O\left( \mathcal{C} \cdot \sum_{i=1}^{d} GR_{KL}(w_i, \mathcal{E}_{avail}) \right) + \max_{e \in \mathcal{E}_{tra}} \mathcal{L}(e, f). \tag{28}$$

Then, summing all kind of inequalities, we can get

$$\sum_{i=1}^{d} \max_{e \in \mathcal{E}_{avail}} \mathcal{L}(e, f) \leq O\left( \mathcal{C} \cdot \sum_{i=1}^{d} GR_{KL}(w_i, \mathcal{E}_{avail}) \right) + \sum_{i=1}^{d} \max_{e \in \mathcal{E}_{tra}} \mathcal{L}(e, f). \tag{29}$$

Hence, averaging these inequalities over d different $g_i \circ w_i$, we obtain

$$\max_{e \in \mathcal{E}_{avail}} \mathcal{L}(e, f) \leq O\left( \mathcal{C} \cdot \frac{1}{d} \sum_{i=1}^{d} GR_{KL}(w_i, \mathcal{E}_{avail}) \right) + \max_{e \in \mathcal{E}_{tra}} \mathcal{L}(e, f). \tag{30}$$

The proof is finished. ∎

