# OpenReview forum: "Generalization Gradient Descent"
_ICLR.cc/2025/Conference — ICLR 2025 Conference Withdrawn Submission_

### Official Review · Reviewer_KCqJ · 2024-10-16

**Soundness:** 2
**Presentation:** 2
**Contribution:** 2
**Rating:** 6
**Confidence:** 2

**Summary:**

This paper introduces a new way of performing gradient descent.  Firstly, this paper defines a certain kind of generalization capability. Then, this paper proposes a method  which selects training sample of each step in a reinforcement learning (RL) manner, thereby improving the gradient descent process.

**Strengths:**

This paper presents a novel approach to gradient descent. It proposes an innovative method that selects the training samples for each step using a RLstrategy, which enhances the overall gradient descent process. This approach demonstrates a creative combination of RL and optimization techniques, potentially leading to improvements in training efficiency and model performance. The integration of RL for sample selection is a unique aspect that sets this work apart from traditional gradient descent methods, which is quite interesting.

**Weaknesses:**

Line 149: eature -> feature

Line 176: The definition of $W^*$ is a little confusing. Do you mean $\forall W^* \in \mathcal{W}^*$ or $\exists W^* \in \mathcal{W}^*$?

Line 505: The font in the image is too small to be readable.



In Definition 3.1, this paper introduced the definition of the Ideal feature matrix function. However, I still don’t have an intuitive sense of it, and it feels like it’s missing some explanation here. I think it would be better if the reasoning behind this definition or references to relevant literature could be provided.

**Questions:**

Theorem 4.2 provides an upper bound of the generalization error. Is the generalization upper bound tight? Additionally, I noticed that the upper bound of the generalization error is composed of both the generalization ratio (GR) and the training error. The optimization in the RL section is performed with respect to GR. However, as the sample size n increases, which part—GR or training error—will dominate the generalization error? If there are some theoretical results regarding this, I believe the results could be more complete.

In the RL section, the paper uses Q-learning to adjust the training set. The state space is the set of G, and the action space is the selection of training samples. However, I’m a bit confused: for a highly complex model, can its state really be represented by a one-dimensional scalar G (Generalization Ratio)? The theory here seems a bit not solid enough.

Overall, I think the idea of this paper is interesting, but I still have these concerns. I would be happy if the authors could address my questions. Thank you.

---

> ### Author Response · Authors · 2024-11-18
>
> Dear Reviewer KCqJ,
>
> Thanks for your feedback.
> >**Line 149: eature -> feature**
> -  Thank you also for your suggestion about the typo in line 149.
>
> >**Line 176: The definition of $W^\text{*}$ is a little confusing.** **Do you mean $\forall W^\text{*} \in$** **$\mathcal{W}^\text{*}$ or** **$\exists W^\text{*} \in$** **$\mathcal{W}^\text{*}$?**
> - The definition of **$W^\text{*}$** means **$\forall W^\text{*} \in$** **$\mathcal{W}^\text{*}$**, but since **$W^\text{*}$** is inherently a member of **$\mathcal{W}^\text{*}$**, it can be simplified to **$W^\text{*} \in$** **$\mathcal{W}^\text{*}$** in line 176.
>
> >**Line 505: The font in the image is too small to be readable.**
> - We sincerely apologize for the inconvenience caused by the image being too small to read in line 505. Due to space constraints in the paper, we had to reduce its size. However, since we used the TIFF format with high DPI, enlarging the image should resolve this issue. We truly appreciate your valuable suggestion and are actively considering moving some of the images to the Appendix.
>
> >**In Definition 3.1, this paper introduced the definition of the Ideal feature matrix function. However, I still don’t have an intuitive sense of it, and it feels like it’s missing some explanation here. I think it would be better if the reasoning behind this definition or references to relevant literature could be provided.**
> - The definition of the ideal feature matrix function in Definition 3.1 is a novel concept we introduced, representing the ultimate goal achieved by the trained model and containing the conditions necessary for perfect generalization ability, so it is difficult to grasp intuitively. There are currently no related references in the literature, but the concept can be more easily understood by deriving Definition 3.1 from Definitions 3.2 and 3.3.
>
> >**Theorem 4.2 provides an upper bound of the generalization error. Is the generalization upper bound tight?**
> - Theorem 4.2 provides an upper bound for the generalization error, with a strict guarantee obtained in Theorem 3.9.
>
> >**I noticed that the upper bound of the generalization error is composed of both the generalization ratio (GR) and the training error. The optimization in the RL section is performed with respect to GR. However, as the sample size n increases, which part—GR or training error—will dominate the generalization error? If there are some theoretical results regarding this, I believe the results could be more complete.**
> - As the sample size $n$ increases, more training errors and selections (actions) will be made for Gradient Descent, which in turn affects the generalization error dominated by $GR$ and the training error. The number of different domain subsets increases, causing the changes in the values of $GR$ and the changes in the values of training error, which subsequently influences the variation in generalization error.
>
> >**In the RL section, the paper uses Q-learning to adjust the training set. The state space is the set of $G$, and the action space is the selection of training samples. However, I’m a bit confused: for a highly complex model, can its state really be represented by a one-dimensional scalar $G$ (Generalization Ratio)?**
> - In the reinforcement learning (RL) section of this paper, the method we adopt does not rely on Q-learning to adjust the training set. Instead, we accumulate past rewards to calculate scores and use the $ε$-greedy algorithm to determine the optimal action, with the chosen action serving as the basis for scoring. The theoretical foundation of this algorithm is based on optimizers; however, in this context, we take the generalization ratio ($GR$) as the optimization objective.
> - In our algorithm, the RL component first computes the $GR$, then optimizes it using the Generalization Inequality in combination with the losses from the training set. These losses are also used as the scoring criterion. Although highly complex models may not be easily represented or handled with a one-dimensional scalar $G$ (generalization ratio), we believe that through rigorous theoretical derivation, the feasibility of this approach can be established.
>
> Thanks also for the suggestions on the typos. Again, we appreciate your comments on our submission. We hope our response can well address your questions. Please kindly let us know if more clarification is needed. Thank you.

---

> > ### Comment · Reviewer_KCqJ · 2024-11-19
> >
> > I mean, when you use reinforcement learning, you usually need to ensure that the model’s state can be **represented** to some extent by the state variables (e.g., GR in this paper). This is the foundation of reinforcement learning theory, as well as the foundation of MP process. I would like to see more explanation on this, or some references. If I have made any mistakes, please feel free to point them out.
> >
> > By the way, did you use an LLM to write the rebuttal? The phrasing “reinforcement learning (RL)” is not very common. Of course, this won’t affect the scoring—it’s just out of personal curiosity.

---

> > > ### Author Response · Authors · 2024-11-19
> > >
> > > Dear Reviewer KCqJ,
> > >
> > > Thanks for your constructive and valuable comments on our paper.
> > >
> > > Firstly, we provide a detailed introduction to MP, followed by an explanation of our process. The Markov Process (MP) [1] consists of $(\mathcal{X}, \mathcal{A}, \mathbb{P}, r, \gamma)$, where $\mathcal{X}$ is a $\textit{continuous state space}$ $\mathcal{X} \subseteq \mathbb{R}^d$, and $\mathcal{A}$ is a $\textit{finite action set}$. We further let $X_t \in \mathcal{X}$ denote the state at time $t$, and $A_t \in \mathcal{A}$ denote the action at time $t$. Then, the measure $\mathbb{P}$ defines the action-dependent transition kernel for the underlying Markov chain $\{X\}_{t,t \geq 0}$:
> > >
> > > $\mathbb{P}(X_{t+1} \in U \mid X_t = x, A_t = a) = \int_U P(dy \mid x, a),$ for any measurable set $U \subseteq \mathcal{X}$.
> > >
> > > The one-stage reward at time $t$ is given by $r(X_t, A_t)$, where $r : \mathcal{X} \times \mathcal{A} \to \mathbb{R}$ is the reward function, and is assumed to be uniformly bounded, i.e., $r(x, a) \in [0, r_{\text{max}}],$ for any $(x, a) \in \mathcal{X} \times \mathcal{A}$. Finally, $\gamma$ denotes the discount factor.
> > >
> > > A stationary policy maps a state $x \in \mathcal{X}$ to a probability distribution $\pi(\cdot \mid x)$ over $\mathcal{A}$, which does not depend on time. For a policy $\pi$, the corresponding value function $V^\pi : \mathcal{X} \to \mathbb{R}$ is defined as the expected total discounted reward obtained by actions executed according to $\pi$:
> > >
> > > $V^\pi(x_0) = \mathbb{E} \left[ \sum_{t=0}^\infty \gamma^t r(X_t, A_t) \mid X_0 = x_0 \right].$
> > >
> > > The action-value function $Q^\pi : \mathcal{X} \times \mathcal{A} \to \mathbb{R}$ is defined as
> > >
> > > $Q^\pi(x, a) = r(x, a) + \gamma \int_{\mathcal{X}} P(dy \mid x, a) V^\pi(y).$
> > >
> > > The goal is to find an optimal policy. **In contrast**, our $V_\pi : \mathbb{R} \to \mathbb{R}$ is defined as the expected total reward obtained by actions executed according to $\pi$:
> > >
> > > $V^\pi(g_0) = \mathbb{E_{\pi}} \left[ \sum_{i=0}^t \mathcal{R}(g_i', a_i') \mid G_0 = g_0 \right].$
> > >
> > > We also define the action-value function $Q^\pi : \mathcal{G}' \times \mathcal{A}' \to \mathbb{R}$ as
> > >
> > > $Q^\pi(g_t', a_t') = V_\pi(g_0) + \mathcal{R}\left(U\left(g_t, g_{t-1} a_t, a_{t-1}\right)\right).$
> > >
> > > Our process differs from traditional MDP. We calculate the reward function $\mathcal{R}$ based on the rewards from past actions, $\mathcal{R}\left(U\left(g_t, g_{t-1} a_t, a_{t-1}\right)\right)$. Additionally, our state space $G \in [0, +\infty]$ and our loss function $\mathcal{L}(e'', f) \in [0, +\infty]$, allowing the current state to be derived from the relationship $(g_t, a_t)$  (Line 390,399) $= f(g_{t-1}, a_{t-1})$.
> > >
> > > However, slight perturbations in $(g_t, a_t)$ can lead to an overly sensitive response, which introduces instability or overfitting risks. Despite these challenges, our experimental results demonstrate that the model effectively captures the relationship between actions and the environment. In summary, $GR$ ensures that the model’s state can be represented by the state variables.
> > >
> > > Furthermore, thank you for your observation. We did not use the LLM to write the rebuttal, the phrasing "reinforcement learning (RL)" is a abbreviation commonly used in numerous academic papers [2, 3]. We appreciate your feedback on the phrasing.
> > > > **REFERENCES**
> > > - [1] Zou, S., Xu, T., & Liang, Y. (2019). Finite-sample analysis for sarsa with linear function approximation. Advances in neural information processing systems, 32.
> > > - [2] Li, Y. (2017). Deep Reinforcement Learning: An Overview. arXiv preprint arXiv:1701.07274.
> > > - [3] Arulkumaran, K., Deisenroth, M. P., Brundage, M., & Bharath, A. A. (2017). Deep reinforcement learning: A brief survey. IEEE Signal Processing Magazine, 34(6), 26-38.
> > >
> > > Please let us know whether these answers have resolved the reviewer's concerns. Again, we appreciate your comments on our submission. We hope our response can well address your questions. Thank you.

---

### Official Review · Reviewer_t6dp · 2024-10-28

**Soundness:** 2
**Presentation:** 1
**Contribution:** 2
**Rating:** 3
**Confidence:** 2

**Summary:**

The paper proposes a variation of the framework of (Ye et al., Neurips 2021) to analyze out-of-distribution generalization, then propose a training algorithm that uses these concepts to offer cross-domain generalization performance.

Ye et al. (2021) studies how to use a small set of domains Eavail to construct a classifier that works across a larger set of domains Eall that we obviously are not expected to know in advance. They provide bounds that depends on an essentially unknowable "expansion function" that measures how features qualities expand from Eavail to Eall.  This paper instead assumes that all domains are available in a set Eavail that is partitioned into training domains Etrain and validation domains Eval. There is no Eall anymore, and one wonder why one should not train using all the known domains of Eavail (which are all the domains that matter) rather than a subset.  Other differences relative to (Ye et al. 2021) include an attempt to talk about features in all network layers (instead of just the penultimate one).

**Strengths:**

This is an ambitious paper.

**Weaknesses:**

* This paper instead assumes that all domains are available in a set Eavail that is partitioned into training domains Etrain and validation domains Eval. There is no Eall anymore, and one wonder why one should not train using all the known domains of Eavail (which are all the domains that matter) rather than a subset.

* There is something disturbing in equation (3) [Thm 2.1].   Remember that W(X) is a dxN matrix.  An expression of the form P(W(X^{e_i})|...) should be normalized over the values taken by this matrix, not over the i indice representing the domains.  Either the notation is inadequate (meaning that the authors mean something else) or the theorem is wrong.

* Through definition 3.8, algorithm GGD depends on all the domain distributions in Eavail and not just the domain distributions in Etra.  Since one wishes to be robust in Eavail  (unlike Ye et al. (2021), there is no Eall here], why not use Eavail directly.  The only justification that comes to my mind would be when Etra is small and Eval is much larger.  But evaluating GR_{KL}(W, Eavail) does not seem cheap.

* I must recognize that I find all this very unclear.

**Questions:**

Please see the weaknesses section.

---

> ### Author Response · Authors · 2024-11-18
>
> Dear Reviewer t6dp,
>
> Thanks for your feedback.
> >**This paper instead assumes that all domains are available in a set $\mathcal{E}_{avail}$** **that is partitioned into training domains $\mathcal{E}_{train}$** **and validation domains $\mathcal{E}_{val}$.** **There is no $\mathcal{E}_{all}$ anymore, and one wonder why one should not train using all the known domains of** **$\mathcal{E}_{avail}$ (which are all the domains that matter) rather than a subset.**
> - We use **$\mathcal{E}_{avail}$** to encompass all known domains. During training, **$\mathcal{E}_{tra}$** is used for performing Gradient Descent, while **$\mathcal{E}_{tra}$** and **$\mathcal{E}_{val}$** are combined to calculate the Generalization Ratio. This means the entire **$\mathcal{E}_{tra}$** is utilized for training, while its subsets are employed for Generalization Ratio computation. Moreover, since the number of **$\mathcal{E}_{val}$** samples extracted from **$\mathcal{E}_{avail}$** is relatively small, applying the GGD algorithm not only helps prevent overfitting but also significantly enhances generalization performance.
>
> >**There is something disturbing in equation (3) [Thm 2.1]. Remember that $W(X)$ is a $d \times N$ matrix. An expression of the form $P(W(X^{e_i})|...)$ should be normalized over the values taken by this matrix, not over the $i$ indice representing the domains. Either the notation is inadequate (meaning that the authors mean something else) or the theorem is wrong.**
> - In equation (3) [Thm 2.1], $W(X)$ represents the probability of a matrix being generated under class $y$. Specifically, it denotes the likelihood of the current $W(X)$ matrix being generated (which differs from the traditional understanding). Therefore, this probability can be computed using conditional probability without the need for normalization over the possible values of the matrix.
>
> >**Through Definition 3.8, algorithm GGD depends on all the domain distributions in **$\mathcal{E}_{avail}$** and not just the domain distributions in **$\mathcal{E}_{tra}$**. Since one wishes to be robust in **$\mathcal{E}_{avail}$** (unlike Ye et al. (2021), there is no **$\mathcal{E}_{all}$** here], why not use Eavail directly. The only justification that comes to my mind would be when **$\mathcal{E}_{tra}$** is small and **$\mathcal{E}_{val}$** is much larger. But evaluating **$GR_{KL}(W, \mathcal{E}_{avail})$** does not seem cheap.**
> - In the GGD algorithm, we use **$\mathcal{E}_{tra}$** only for Gradient Descent, while **$\mathcal{E}_{val}$** plays a role of a validation set in the calculation of the Generalization Ratio. Therefore, the proportion of **$\mathcal{E}_{val}$** is sampled based on the traditional machine learning ratio, resulting in a relatively small quantity. As a result, we directly use **$\mathcal{E}_{avail}$** during training and later incorporate the above concept to calculate the Generalization Ratio.
>
> >**I must recognize that I find all this very unclear.**
> - We hope our response above effectively addresses your questions.
>
> Thank you also for the suggestions in our paper.  Again, we appreciate your comments on our submission. Please kindly let us know if more clarification is needed. Thank you.

---

> > ### Comment · Reviewer_t6dp · 2024-11-25
> >
> > I have read your responses but I did not get an answer to my main concern.
> >
> > * If all the possible environments are available at training time, the problem becomes a distributional robust optimization problem for which there are lots of valid approaches and also a lot of limitations.
> >
> > * If we do not assume that all the environments of interest are available at training time, one can use some of them as a validation set to try to control how the system behaves on unavailable environment. Without Ye's expansion function, this implicitly relies on the assumption that the available environments are a "random sample" of the actual environments, so that a good performance on the validation environment is a good indication of the performance on all environments. But then the n that appears in that laws of large number is not the number of examples, but the number of environments (which is much smaller.)

---

> > > ### Author Response · Authors · 2024-11-25
> > >
> > > Dear Reviewer t6dp,
> > >
> > > Thanks for your feedback.
> > >
> > > > **If all the possible environments are available at training time, the problem becomes a distributional robust optimization problem for which there are lots of valid approaches and also a lot of limitations.**
> > > - Thank you for your valuable feedback. We would like to clarify that while our work incorporates concepts that might appear related to Distributed Robust Optimization (DRO), our approach is fundamentally rooted in the framework of reinforcement learning and is distinct from DRO. Specifically, DRO focuses on designing robust solutions for static optimization problems over uncertainty sets, whereas our method emphasizes dynamically interacting with the environment to learn policies that maximize cumulative performance. Additionally, our algorithm does not assume access to all environments but leverages exploration mechanism from reinforcement learning combined with expansion function (Ye et al. (2021)) to effectively handle discrepancies between training and validation environments. This allows our approach to be more flexible and dynamically adaptive when managing behavior in unseen environments. We hope this clarification addresses the misunderstanding and highlights the uniqueness of our method.
> > >
> > > > **If we do not assume that all the environments of interest are available at training time, one can use some of them as a validation set to try to control how the system behaves on unavailable environment. Without Ye's expansion function, this implicitly relies on the assumption that the available environments are a "random sample" of the actual environments, so that a good performance on the validation environment is a good indication of the performance on all environments. But then the n that appears in that laws of large number is not the number of examples, but the number of environments (which is much smaller.)**
> > > - We agree that traditional approaches implicitly rely on the assumption that available environments form a representative "random sample" of all possible environments. However, as you noted, this assumption may become unreliable when the number of environments is limited. Our approach, grounded in expansion function (Ye et al. (2021)), specifically addresses this issue by reducing dependence on this assumption through a principled mathematical framework.
> > >
> > > - Our method leverages the expansion function to model and control the behavior of the system across unavailable environments, even when the available environments are limited in number. The key mechanisms are as follows:
> > >
> > > 1. **Minimizing Variation Across Environments**:
> > >    By optimizing the variation term **$\mathcal{V}_{KL}(W, $** **$\mathcal{E}_{avail})$**, our approach ensures that the behavior of model is consistent across the environments it encounters. This reduces reliance on the assumption that the validation environments perfectly represent all environments.
> > >
> > > 2. **Maximizing Informativeness**:
> > >    The expansion function incorporates the informativeness term **$\mathcal{I}_{KL}(W, $** **$\mathcal{E}_{tra})$**, which ensures that the training environments provide sufficient information to generalize to unseen environments, regardless of their representativeness as a "random sample."
> > >
> > > 3. **Beyond Empirical Sampling**:
> > >    Rather than relying solely on the empirical sampling of environments, the expansion function uses a theoretical framework to link available environments to the broader set of potential environments. This allows our method to extrapolate performance to unavailable environments without requiring a large number of available environments.
> > >
> > > - We acknowledge that in our framework, the "sample size" $n$ corresponds to the number of environments rather than the number of data examples. However, our method does not rely on large $n$ for statistical guarantees. Instead, it achieves generalization by formalizing and minimizing the worst-case variation between environments, ensuring robustness even when $n$ is small. This makes our approach particularly effective in scenarios where the number of environments is inherently limited.
> > >
> > > - Our experiments demonstrate that the proposed method achieves robust generalization across diverse OOD benchmarks, even when the number of available environments is small. This empirical evidence supports the claim that our framework effectively mitigates the limitations of the random sample assumption. While traditional methods depend on a strong assumption about the representativeness of available environments, our method introduces a principled expansion function that minimizes this dependency. By explicitly modeling variation and informativeness, we ensure reliable performance across unseen environments, even when the number of available environments is limited.
> > >
> > > We hope this addresses your concern and clarifies how our approach overcomes this limitation. We welcome further feedback to refine and strengthen our work.

---

> > > > ### Comment · Reviewer_t6dp · 2024-11-25
> > > >
> > > > Reading your response makes me believe I missed something important in the paper. I will try to make the time to re-read carefully but I am not sure I'll be able to do before the end of the discussion period. In the meantime, I have reduced the confidence of my assessment to reflect my worry.
> > > >
> > > > That said, I am also a good example of the kind people most likely to read your paper. The fact that I might have misread it so badly is in itself a bit of a concern. It might be useful to start by explaining (or previewing) how your approach to the o.o.d. problem differs from earlier approaches that people might know, maybe by using some of the language in your latest reply.

---

> > > > > ### Author Response · Authors · 2024-11-26
> > > > >
> > > > > Dear Reviewer t6dp,
> > > > >
> > > > > Thanks for your feedback.
> > > > >
> > > > > We sincerely appreciate your thoughtful feedback and the time you have taken to engage with our work. We understand your concerns and regret any confusion caused by our presentation. To address this constructively, we will clarify the novelty and contributions of our approach, particularly in tackling the out-of-distribution (OOD) problem. Specifically, we will highlight the generalization ratio ($GR$) as a novel metric for evaluating OOD performance, explain how our Generalization Gradient Descent (GGD) algorithm formalizes and optimizes the trade-off between in-distribution and out-of-distribution loss (an advancement over traditional gradient descent methods) and connect our work to key earlier approaches, explicitly contrasting our methodology and results.
> > > > >
> > > > > Again, we appreciate your comments on our submission. We hope our response can well address your questions. Please kindly let us know if more clarification is needed. Thank you.

---

> > > > > > ### Author Response · Authors · 2024-12-02
> > > > > >
> > > > > > Dear Reviewer t6dp,
> > > > > >
> > > > > > Since the discussion period draws to a close in the next seven hours, we were wondering if you have had a chance to go through our responses. Please let us know if your questions are addressed, we are happy to clarify anything remaining or any new questions. Thank you very much! Again, we appreciate your comments on our submission.

---

### Official Review · Reviewer_fd14 · 2024-11-01

**Soundness:** 3
**Presentation:** 3
**Contribution:** 3
**Rating:** 6
**Confidence:** 2

**Summary:**

The paper tackles the out-of-distribution (OOD) generalization problem. The main ideas are to consider the distribution of features invariant to domain shifts and to formulate them as the ideal feature matrix function, generalized model, and non-generalized model.  The paper also introduces the variation, the informativeness, and the generalization decision process (GDP) to propose a new algorithm called the generalization gradient descent (GGD) based on the framework of reinforcement learning. The GGD performs better than the traditional gradient descent in their experiment.

**Strengths:**

The paper discusses the generalization problem mathematically rigorously. The ideas of generalization ratio and generalization decision process are novel and the introduction of reinforcement learning framework is unique.

**Weaknesses:**

It is not clear whether the framework is practical or not since the definition of the ideal feature matrix function that the posterior distributions of y are the same seems too strong to exist in real problems. The reviewer recommends showing some examples. The experiment is rather small and specific, that is, it compares only GGD and TGD. The effectiveness of the proposed algorithm should have been confirmed with a larger number of problems. In addition, this work strongly depends on the previous work (Ye+ 2021) and its difference seems not large since many of the concepts were introduced there.

**Questions:**

None.

---

> ### Author Response · Authors · 2024-11-18
>
> Dear Reviewer fd14,
>
> Thanks for your feedback. First and foremost, we sincerely appreciate your thoughtful feedback and concerns regarding this paper. Allow us to provide a more detailed explanation of the framework.
>
> > **It is not clear whether the framework is practical or not since the definition of the ideal feature matrix function that the posterior distributions of y are the same seems too strong to exist in real problems. The reviewer recommends showing some examples.**
> - The definition of an ideal feature matrix function with identical posterior distributions of $y$ is intended to serve as a benchmark for the theoretical analysis of the algorithm in this paper, rather than as a requirement to be fully satisfied in all situations. This idealized assumption provides researchers with a clear objective, enabling them to measure the gap between real-world feature matrix functions and the ideal state, thereby improving algorithm performance.
>
> > **The experiment is rather small and specific, that is, it compares only GGD and TGD. The effectiveness of the proposed algorithm should have been confirmed with a larger number of problems. In addition, this work strongly depends on the previous work (Ye+ 2021) and its difference seems not large since many of the concepts were introduced there.**
> - The work by Ye et al. (2021) primarily focuses on the theoretical foundation of OOD (Out-of-Distribution) generalization, with its experimental algorithm section limited to discussing domain accuracy in conjunction with **$\mathcal{V}_{KL}$**. In contrast, our study extends the theoretical framework proposed by Ye et al. (2021) by introducing the generalization ratio ($GR$) and integrating it with the Generalization Bound (Ye et al. (2021)). Furthermore, we incorporate reinforcement learning (RL) to develop the GGD algorithm, bridging the gap between abstract theory and practical application.
>
> - Through this process, we observed that the purely mathematical OOD theory is inherently complex, making it susceptible to the limitations of its mathematical assumptions when applied in practice. To address this, we have simplified the OOD theoretical framework, emphasizing the extraction of key theoretical elements for practical algorithmic implementation. Additionally, we have found that using RL as a bridge effectively connects OOD theory with real-world algorithmic performance, yielding optimal results. While our experiments are relatively modest in scale and scope, we are confident that this work represents a meaningful and innovative contribution.
>
> Again, we appreciate your comments on our submission. We hope our response can well address your questions. Please kindly let us know if more clarification is needed. Thank you.

---

> > ### Comment · Reviewer_fd14 · 2024-11-22
> >
> > Thanks for your response. I raised the evaluation a bit.

---

> > > ### Author Response · Authors · 2024-11-22
> > >
> > > Thank you for your valuable feedback and comments on our work. We sincerely appreciate your comments and the time you spend reviewing our submissions.

---

### Official Review · Reviewer_EyVK · 2024-11-02

**Soundness:** 3
**Presentation:** 2
**Contribution:** 3
**Rating:** 6
**Confidence:** 2

**Summary:**

This paper proposes a novel framework for analyzing out-of-distribution (OOD) generalization, integrating a generalization ratio to quantify generalization capacity and a generalization decision process (GDP) to link losses across seen and unseen domains. This framework enables backpropagation-based training without model selection, using a combination of informativeness and variation in generalization to derive a generalization inequality, with experiments showcasing its effectiveness in improving generalization ability.

**Strengths:**

1. They designed an algorithm focused on enhancing the generalization ability of gradient descent, leveraging the concept of a generalization ratio and insights from reinforcement learning.
2. They experimentally demonstrate that the algorithm outperforms traditional gradient descent.

**Weaknesses:**

1. The title is intriguing. Would it be clearer to use 'Gradient Descent with the Generalization Ratio: Enhancing.......' instead?
2. In my view, GGD is proposed based on reinforcement learning and the generalization ratio. However, the motivation behind the algorithm design could be clarified further.
3. There is considerable overlap with the paper by Ye et al. (2021). It might help to elaborate more on the ideas behind the algorithm's design.
4. The writing could be improved, as there are several grammatical errors throughout.

**Questions:**

1. You introduced the concept of the generalization ratio (GR) in this paper. Could you clarify the functional difference between V_KL for different e_i sets and GR?
2. As a novel and important criterion, could you elaborate on the properties of GR and why it serves as a strong evaluation metric?
3. Could you design some experiments that demonstrate the advantages of using GR instead of V_KL as Ye et al. (2021) also provided a model selection criterion?

---

> ### Author Response · Authors · 2024-11-18
>
> Dear Reviewer EyVK,
>
> Thanks for your feedback.
> > **The title is intriguing. Would it be clearer to use 'Gradient Descent with the Generalization Ratio: Enhancing.......' instead?**
> - We chose **Generalization Gradient Descent** as the title mainly to combine **Generalization** and **Gradient Descent**, presenting a novel and intriguing algorithm concept. However, if clarity is the priority, a title like "Gradient Descent with the Generalization Ratio: Enhancing..." might be more straightforward in conveying the focus of the research. Do you prefer keeping the original creative approach, or would you lean towards a clearer description?
>
> > **In my view, GGD is proposed based on reinforcement learning and the generalization ratio. However, the motivation behind the algorithm design could be clarified further. There is considerable overlap with the paper by Ye et al. (2021). It might help to elaborate more on the ideas behind the algorithm's design.**
> - Initially, we attempted to combine the Generalization Bound from Ye et al. (2021) with Gradient Descent. However, we found that deriving the Expansion Function (Definition 3.3 in Ye et al. (2021)) was highly challenging. As a result, we turned to the Generalization Ratio ( $GR$ ) to provide a more robust guarantee. Subsequently, we discovered that directly applying the theory from Ye et al. (2021) with Gradient Descent led to increased errors. On the other hand, integrating reinforcement learning actions with the Generalization Bound from Ye et al. (2021) yielded significantly better results. This process formed the basis for designing the GGD algorithm.
>
> > **The writing could be improved, as there are several grammatical errors throughout.**
> - Thanks for the suggestions on the grammatical errors in our paper.
>
> > **You introduced the concept of the generalization ratio ( $GR$ ) in this paper. Could you clarify the functional difference between $\mathcal{V}_{KL}$ for different $e_i$ sets and $GR$ ?**
> - In our paper, we define the **$\mathcal{V}_{KL}$** function to compute scalar mappings from two different domains. We divide one domain **$\mathcal{E}_{\text{tra}}$** into $n''$ subsets $e_i''$, as the number of actions in the GGD algorithm is related to the number of $e_i''$ subsets. When the number of $e_i''$ subsets is smaller, the available actions are fewer, leading to smaller differences in the impact of choices and relatively smoother variations. Conversely, when the number of $e_i''$ subsets is larger, the available actions increase, resulting in greater variations. For the other domain, **$\mathcal{E}_{\text{val}}$**, due to its relatively smaller sample size, it is divided into the size of subset $e_i'$, resulting in different numbers of subsets between the two domains (Line 383). Moreover, the $GR$ function is built on the **$\mathcal{V}_{KL}$** function and further regularized through the $\mathcal{I}_{KL}$ function to ensure the generalization ability for OOD tasks (Theorem 3.9).
>
> >**As a novel and important criterion, could you elaborate on the properties of $GR$ and why it serves as a strong evaluation metric?**
> - The core concept of $GR$ is to quantify generalized states, ensuring that minimizing variation does not compromise the importance of maintaining a certain level of informativeness. This concept is rigorously established within a mathematical framework, addressing the limitations of Ye et al. (2021), where **Definition 3.3** only provides a qualitative description of OOD.
>
> >**Could you design some experiments that demonstrate the advantages of using $GR$ instead of $\mathcal{V}_{KL}$ as Ye et al. (2021) also provided a model selection criterion?**
> - $GR$ is established on a rigorous mathematical framework, and its values are closely related to the accuracies across different domains. Since the scope of domains considered by $GR$ has been expanded to **$\mathcal{E}_{\text{all}}$** through Theorem 3.9, we build $GR$ on the theoretical final results and use the accuracies from different domains to evaluate its advantages. Additionally, the training dynamics shown in Figure 1 are used solely to illustrate trends. In contrast, the model selection criteria proposed by Ye et al. (2021) rely solely on the **$\mathcal{V}_{KL}$** function as the foundation of their algorithm. The $GR$ used in the GGD algorithm not only incorporates the Generalization Inequality (Theorem 4.2) but also integrates reinforcement learning, making it more comprehensive.
>
> Thanks also for the suggestions on the grammatical errors in our paper. Again, we appreciate your comments on our submission. We hope our response can well address your questions. Please kindly let us know if more clarification is needed. Thank you.

---

> > ### Comment · Reviewer_EyVK · 2024-11-26
> >
> > Thanks so much for your reply! I'll raise my score.

---

> > > ### Author Response · Authors · 2024-11-26
> > >
> > > Thank you for your valuable feedback and comments on our work. We sincerely appreciate your comments and the time you spend reviewing our submissions.

---

### Author Response · Authors · 2024-11-26

We have submitted the revised version of our paper with corrections to the typos. We would like to extend our sincere gratitude to all the reviewers for their valuable feedback. We remain confident in the strengths of this submission, which are primarily grounded in the OOD mathematical framework proposed by Ye et al. (2021). In our work, we demonstrate innovative, simple, meaningful, and scalable algorithms that perform comparably or even surpass traditional gradient descent methods. These conventional approaches are often more prone to overfitting, unable to handle OOD scenarios effectively, and lack groundbreaking advancements. We believe that we have addressed all of the questions posed by the reviewers. We would appreciate if the reviewers could update their evaluations and scores, clearly articulating any aspect they still don't like about the paper so that we can continue improving it for the future.

---

### Note · Authors · 2025-01-23

I have read and agree with the venue's withdrawal policy on behalf of myself and my co-authors.